# Long decay length of magnon-polarons in BiFeO$_3$/La$_{0.67}$Sr$_{0.33}$MnO$_3$ heterostructures

Jianyu Zhang[1,13], Mingfeng Chen [2,13], Jilei Chen[1,3,13], Kei Yamamoto [4,5,13], Hanchen Wang[1,13], Mohammad Hamdi [6,13], Yuanwei Sun[7], Kai Wagner[8], Wenqing He[9], Yu Zhang [9], Ji Ma[2], Peng Gao [7], Xiufeng Han [9], Dapeng Yu[3], Patrick Maletinsky [8], Jean-Philippe Ansermet[10], Sadamichi Maekawa[5,11✉], Dirk Grundler [6,12✉], Ce-Wen Nan [2✉] & Haiming Yu [1✉]

Magnons can transfer information in metals and insulators without Joule heating, and therefore are promising for low-power computation. The on-chip magnonics however suffers from high losses due to limited magnon decay length. In metallic thin films, it is typically on the tens of micrometre length scale. Here, we demonstrate an ultra-long magnon decay length of up to one millimetre in multiferroic/ferromagnetic BiFeO$_3$(BFO)/La$_{0.67}$Sr$_{0.33}$MnO$_3$(LSMO) heterostructures at room temperature. This decay length is attributed to a magnon-phonon hybridization and is more than two orders of magnitude longer than that of bare metallic LSMO. The long-distance modes have high group velocities of 2.5 km s$^{-1}$ as detected by time-resolved Brillouin light scattering. Numerical simulations suggest that magnetoelastic coupling via the BFO/LSMO interface hybridizes phonons in BFO with magnons in LSMO to form magnon-polarons. Our results provide a solution to the long-standing issue on magnon decay lengths in metallic magnets and advance the bourgeoning field of hybrid magnonics.

[1] Fert Beijing Institute, MIIT Key Laboratory of Spintronics, School of Integrated Circuit Science and Engineering, Beihang University, Beijing, China. [2] State Key Laboratory of New Ceramics and Fine Processing, School of Materials Science and Engineering, Tsinghua University, Beijing, China. [3] Shenzhen Institute for Quantum Science and Engineering, Southern University of Science and Technology, Shenzhen, China. [4] Advanced Science Research Center, Japan Atomic Energy Agency, Tokai, Ibaraki 319-1195, Japan. [5] RIKEN Center for Emergent Matter Science, Wako, Saitama, Japan. [6] Laboratory of Nanoscale Magnetic Materials and Magnonics, Institute of Materials, École Polytechnique Fédérale de Lausanne (EPFL), Lausanne, Switzerland. [7] Interdisciplinary Institute of Light-Element Quantum Materials and Research Center for Light-Element Advanced Materials and International Center for Quantum Materials and Electron Microscopy Laboratory, School of Physics, Peking University, Beijing, China. [8] Department of Physics, University of Basel, Basel, Switzerland. [9] Beijing National Laboratory for Condensed Matter Physics, Institute of Physics, University of Chinese Academy of Sciences, Chinese Academy of Sciences, Beijing, China. [10] Institute of Physics, École Polytechnique Fédérale de Lausanne (EPFL), Lausanne, Switzerland. [11] Kavli Institute for Theoretical Sciences, University of Chinese Academy of Sciences, Beijing, China. [12] Institute of Electrical and Micro Engineering, École Polytechnique Fédérale de Lausanne (EPFL), Lausanne, Switzerland. [13] These authors contributed equally: Jianyu Zhang, Mingfeng Chen, Jilei Chen, Kei Yamamoto, Hanchen Wang, Mohammad Hamdi. ✉email: sadamichi.maekawa@riken.jp; dirk.grundler@epfl.ch; cwnan@mail.tsinghua.edu.cn; haiming.yu@buaa.edu.cn

Spin waves[1,2], the collective excitation of electron spin precession, can propagate over certain distances in magnetic systems. Magnons[3], the quasi-particle representation of spin waves, can convey spin current[4] in ferromagnetic metals[5] and magnetic insulators[6] free of Joule heating and are therefore promising for applications in low-power consumption magnonic devices[7]. Spin waves in bulk insulating yttrium iron garnet (YIG) show record-low magnetic damping. Consistently, spin waves in thin-film YIG exhibit decay lengths of up to a few 100 µm depending on the specific thickness[8–10]. The half-metallic ferromagnetic oxide $La_{0.7}Sr_{0.3}MnO_3$ (LSMO) has recently been used for spin-wave studies[11] owing to its relatively low damping[12], comparable to that of ultra-thin YIG. Typical ferromagnetic metal such as Ni has even larger damping. Ferromagnetic resonance can be driven by surface acoustic waves (SAWs) via magnon–phonon coupling in a thin magnetic film on a piezoelectric substrate[13]. In the coupling process, phonon may carry spin information[14]. Nonreciprocal propagation of SAWs[15–19] is generated by the magnon–phonon coupling. Long-range transfer of angular momentum between two YIG layers separated by a thick substrate[20] is realized by the magnon–phonon coupling[21]. Direct imaging and quantification of both standing and propagating acoustomagnetic waves were presented in hybrid structures consisting of a piezoelectric substrate underneath Ni[22]. This way, magnon–phonon coupling and excitation by a large-area inter-digital transducer (IDT) allowed to transport spin information over a macroscopic distance at specific IDT frequencies[22]. For future hybrid magnonics[23] with multi-frequency operational modes, it is a key to avoid such specific substrate materials and be able to excite propagating magnons directly in all-magnetic material systems, as then much more compact transmission lines and coplanar waveguides can be exploited for excitation of short-wave magnons over broad frequency regimes. Magnetically controlled SAW propagation[24] has been studied in multiferroic $BiFeO_3$ (BFO)[25] exhibiting antiferromagnetic textures[26] and terahertz spin dynamics[27,28]. The BFO/LSMO heterostructure (Fig. 1a) has been intensively studied regarding its exchange bias[29] and the magnetoelectric coupling[30,31]. In this work, we experimentally study the propagation of spin waves in the BFO/LSMO heterostructure (Fig. 1b) using both angle-resolved propagating spin-wave spectroscopy (AR-PSWS) (Fig. 1c, d) and time-resolved Brillouin light scattering (TR-BLS). Two distinct spin-wave modes are observed in the AR-PSWS spectra (Fig. 1d): Mode Y is a coherently propagating spin-wave mode in LSMO that we observe to decay exponentially with a decay length of about 5 µm as discussed later. Mode X (Fig. 1d) is not commonly observed, and we identify it as a hybridized magnon–phonon mode (or magnon-polarons) in the BFO/LSMO heterostructure, which propagates with a high group velocity and long decay length of up to 1 mm. The group velocity of $2.5\,km\,s^{-1}$ observed by TR-BLS suggests that it associates with the BFO phonon mode[24,32]. The assumed hybridization of a phonon mode in BFO with magnons in LSMO and its angular dependency are substantiated by a theoretical approach considering the magnon–phonon coupling[33–36] at the BFO/LSMO interface which possesses both ferromagnetism and magnetoelastic coupling simultaneously (Fig. 1b).

**Spin-wave propagation in BFO/LSMO heterostructures**. The BFO/LSMO bilayer is grown on a $NdGaO_3$ (NGO) substrate. The transmission electron microscope (TEM) characterization of the BFO/LSMO interface is shown in the inset of Fig. 1a (see the "Methods" section). The BFO and LSMO layers are 20- and 80-nm thick, respectively, and characterized by energy-dispersive X-ray spectroscopy from selective elements (Supplementary Fig. 1). To study spin-wave propagation in this material system,

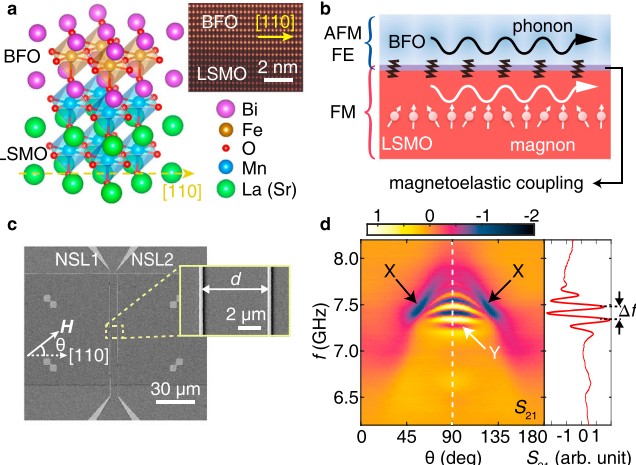

**Fig. 1 BFO/LSMO heterostructures and spin-wave transport. a** A schematic diagram of the interface of the BiFeO3(BFO)/La0.7Sr0.3MnO3(LSMO) heterostructure. The inset shows the atomically resolved interface of BFO/LSMO by annular dark field image. The yellow arrow indicates the pseudo-cubic [110] crystalline orientation of LSMO and BFO. **b** A schematic illustration of magnon–phonon coupling in the BFO/LSMO heterostructure. The system consists of two layers: The top layer BFO is multiferroic being both antiferromagnetic (AFM) and ferroelectric (FE) and hosts a longitudinal acoustic (LA) wave mode. The bottom layer consists of the ferromagnetic (FM) LSMO layer with an in-plane uniaxial anisotropy. The phonons in BFO and magnons in LSMO are hybridized via magnetoelastic coupling at the interfacial layer owing to the interfacial exchange coupling between LSMO and BFO and magnetoelastic interaction within the BFO. **c** A scanning electron microscope (SEM) image of a magnonic device for measuring spin-wave transmission. The field is applied in the plane and rotated with respect to the [110] crystalline orientation of BFO (LSMO). The inset is a zoom-in image showing two integrated gold antennas NSL1 and NSL2 with a center-to-center distance $d = 6$ µm. **d** Spin-wave transmission spectra $S_{21}$ (spin waves excited at NSL1 and detected at NSL2), where mode X and Y are marked. A single spectrum is extracted at θ = 90°, from which a frequency span $\triangle f$ is extracted for the derivation of the spin-wave group velocity used in Eq. (1). The data displayed in the map is the imaginary part of the transmission spectra, and the color bar denotes the amplitude of the imaginary part, which is in arbitrary units.

two nano-stripline (NSL) antennas[37] are integrated on top of the BFO (Fig. 1b) providing excitation over a broadband wavevector regime (Supplementary Fig. 2). The scanning electron microscope (SEM) image of Sample Z02 (see Supplementary Table 1) is shown in Fig. 1c, where the in-plane applied field angle θ is defined with respect to the LSMO [110] orientation as indicated in Fig. 1a. A magnified SEM image of the excitation/detection antennas is shown where the spin-wave propagation distance $d$ is characterized to be 6 µm and the antenna width is ~270 nm for this sample (Supplementary Fig. 2). The ferromagnetic resonance (FMR) of a bare 80-nm-thick LSMO film is measured by a Physical Property Measurement System (PPMS) with a PhaseFMR module (see "Methods"). The intrinsic Gilbert damping $α = 1.2×10^{-3}$ is obtained by the linear fit of the FMR linewidths as a function of resonance frequency (Supplementary Fig. 3). The damping parameter measured in this work is about twice the lowest value of $0.5×10^{-3}$ reported for a 45-nm-thick LSMO film[12]. Figure 1d shows the angle-dependent spin-wave transmission spectra $S_{21}$ (spin waves excited at NSL1 and detected at NSL2) measured by AR-PSWS with a rotating field of 100 mT (see "Methods"). The pair of modes showing a strong absorption with field angle are denoted as mode X (black arrows) and the

one with pronounced signal oscillations is labeled as mode Y (white arrow). Propagating spin waves are known to give rise to signal oscillations due to frequency-dependent phase accumulation along the propagating path[5,8]. The mode X does not show an oscillatory signal variation. This phase-insensitive behavior might be due to the coupling with phonons since the PSWS technique seems to be incompetent in detecting phonon phase information even with IDTs[19,38]. On the bare reference LSMO sample without BFO, mode X is absent but mode Y is observed clearly (Supplementary Fig. 4). For mode Y, at θ = 90° (field applied perpendicular to the spin-wave wavevector) i.e., in the Damon-Eshbach (DE) configuration, a lineplot is extracted and shown in the right panel of Fig. 1d, where a frequency span $\triangle f$ is indicated corresponding to a phase oscillation of 2π detected by the AR-PSWS.

**Angle-dependent spin-wave transmission: experiments and simulations**. The coherent spin waves excited by NSL antennas have a non-zero wavevector distribution $\triangle k$, and thus the propagation over a certain distance $d$ yields a phase difference of $\triangle \varphi = \triangle k \cdot d$. Since the AR-PSWS based on vector network analyzer is phase sensitive, a corresponding signal oscillation of period $\triangle f$ (right panel of Fig. 1d) is observed when $\triangle \varphi$ reaches 2π and multiples thereof. Therefore, the spin-wave group velocity can be calculated using[5,8]

$$v_{\mathrm{g}} = \frac{\partial \omega}{\partial k} \approx \frac{2\pi \triangle f}{2\pi/d} = \triangle f \cdot d \quad (1)$$

where $d$ is the spin-wave propagation distance as shown in Fig. 1c. The evolution of mode Y as a function of propagation distance is presented in Fig. 2a–e. The phase oscillation of mode Y becomes more rapid, i.e., $\triangle f$ decreases when $d$ increases in agreement with Eq. (1). The spin-wave group velocity $v_{\mathrm{g}}$ can then be extrapolated from a linear fitting of $\triangle f$ versus $1/d$ as shown in Fig. 2i, where an average value is obtained for mode Y, which is consistent with the thin-film approximation of the DE spin-wave mode[39].

To understand the results of AR-PSWS and in particular the unusual characteristics of mode X, we studied a theoretical

model[16] where a thin-film spin-wave mode couples with a longitudinal acoustic (LA) wave mode via isotropic magnetoelastic coupling. The material parameters of the model include the mass density $\rho = 8400 \ \mathrm{kg \ m^{-3}}$ and LA sound velocity $c_{\mathrm{s}} = 2.5 \ \mathrm{km \ s^{-1}}$ of BFO, an in-plane uniaxial anisotropy $K = 4 \ \mathrm{kJ \ m^{-3}}$ along with thickness $t = 80 \ \mathrm{nm}$, saturation magnetization $M_{\mathrm{s}} = 276 \ \mathrm{kA \ m^{-1}}$ for LSMO extracted from the FMR spectrum[40], and an effective magnetoelastic coupling $b_{\mathrm{eff}} = 2 \times 10^5 \ \mathrm{J \ m^{-3}}$. The detailed discussion on the coupling is contained in Supplementary Sec. V. We note that the value $b_{\mathrm{eff}}$ is quite large compared to bare LSMO and can be presumably accounted for as the strong magneto-striction of the BFO layer[41]. The simulation of AR-PSWS measurements based on this model (Fig. 2f–h) correctly reproduces the dependence of $S_{21}$ on the magnetic field angle, excitation frequency, and the distance between the antennas (see Supplementary Fig. 7 for more simulation results). Mode X is thus identified with the magnon–phonon hybridization induced by the interface coupling.

The series of angle-dependent measurements on Samples Z02–Z06 with a differently oriented wavevector $k$ (see Supplementary Fig. 11) reveals an existing uniaxial magnetic anisotropy along the LSMO [110] crystal direction (Fig. 1a). This anisotropy results in aligned magnetic domain stripes along the LSMO [110] direction, no matter in which direction the film is initially saturated, as characterized by scanning nitrogen-vacancy (NV) magnetometry (see "Methods"). The direct comparison of NV measurements on samples with and without BFO on top of LSMO suggests that this uniaxial anisotropy[42] might be associated with the BFO/LSMO interface (Supplementary Fig. 12). The resonance frequency of the mode X is found to be not very sensitive to the LSMO thickness, which implies that it cannot correspond to perpendicular standing spin waves[43–45].

**High group velocities of the magnon–phonon hybridized mode**. If the NSL antennas are replaced by coplanar waveguide

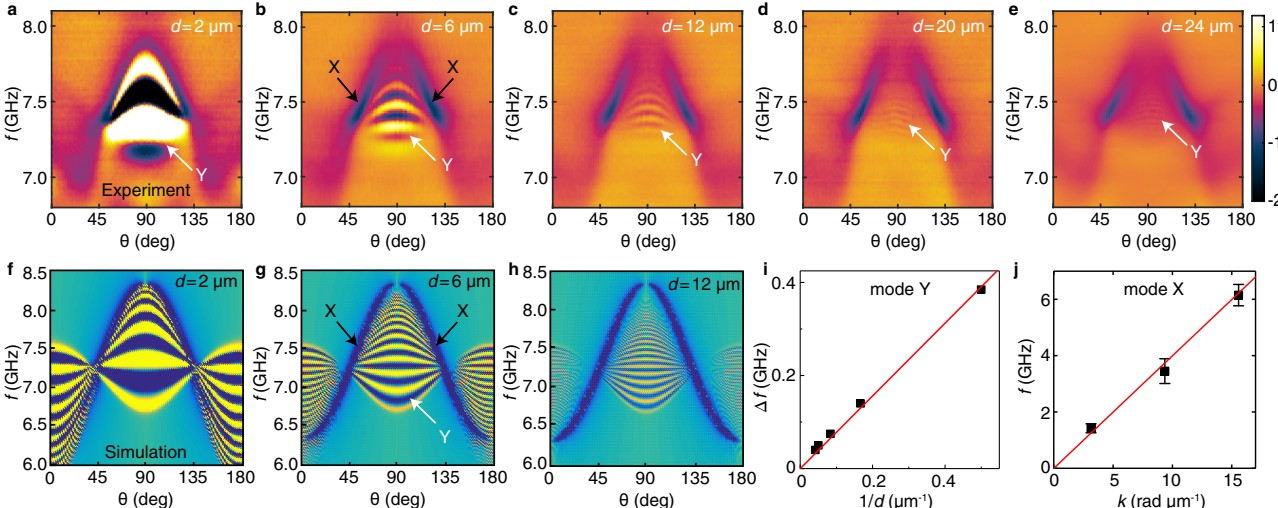

**Fig. 2 Distance-dependent transmission spectra and spin-wave group velocities. a–e** Transmission spectra $S_{21}$ measured as a function of field angle θ from 0 to180 degree on Samples Z01, Z02, Z07, Z08, and Z09 with spin-wave propagation distances $d$ of 2, 6, 12, 20, and 24 μm, respectively. The spin-wave wavevector $k$ is along the LSMO [110] direction. The data displayed in the map is the imaginary part of the transmission spectra, and the color bar denotes the amplitude of the imaginary part, which is in arbitrary units. **f–h** Simulation results of angle-dependent spin-wave transmission over 2, 6, and 12 μm. The color code is taken from the real part of the total transmission signal $s(d)$ picked by the NSL or CPW in the simulations (see Supplementary Sec. V). **i** Peak-to-peak frequency span $\triangle f$ extracted from signals attributed to mode Y in **a–e** at θ = 90° and plotted as a function of $1/d$. The linear fit (red line) yields an average group velocity of 0.8 km s⁻¹. **j** The dispersion relation obtained by frequencies $f$ of mode X extracted from the field-dependent transmission spectra at 54° (Supplementary Fig. 11) and wavevectors $k$ from the Fourier transformation of CPWs (Supplementary Fig. 2).

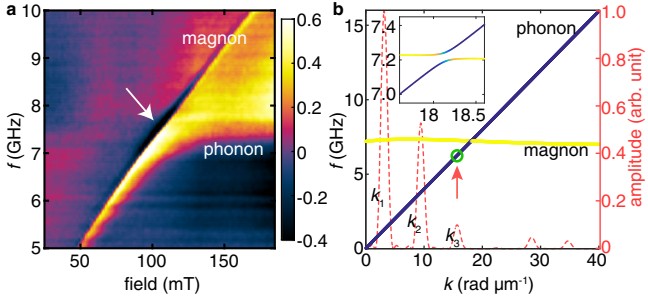

**Fig. 3 The mode hybridization induced by magnon–phonon coupling.**
**a** Transmission amplitude $S_{21}$ measured on Sample Z15 as a function of magnetic field applied in the direction of θ = 54°. An anticrossing-like feature is observed between the magnon mode and the phonon mode. The field-independent phonon mode is attributed to the high-order ($k_3$) excitation from the CPW antenna. The white arrow marks the magnon–phonon hybridized mode (mode X) with strong intensity. The data displayed in the color map is the imaginary part of the transmission spectra, and the color bar denotes the amplitude of the imaginary part, which is in arbitrary units. The data are treated with a background subtraction of the reference spectrum at zero field. **b** Simulation results on the hybridization of the magnon and phonon dispersions with a fixed applied field of 100 mT at θ = 45°. The blue and yellow lines stand for the phonon and magnon branches, respectively. The zoom-in dispersion around the crossing point is shown in the inset where the green color suggests the magnon–phonon hybridization. The wavevector ($k$) excitation profile (red dashed line) is calculated by Fourier transformation of the CPW antenna. The amplitude (scales on the right) is normalized to that of the $k_1$ excitation. The green circle (at $k_3$ excitation) indicates the hybridization off the crossing point that may lead to the anticrossing-like behavior in (**a**).

(CPW) antennas[5,8,46] (e.g., Sample Z15), the excitation is known to exhibit a specific wavevector distribution with pronounced discrete peaks (red dashed line in Fig. 3b and Supplementary Fig. 2). In the field-dependent transmission spectra measured on Sample Z15, three distinct modes exist that are independent of the magnetic field (Supplementary Fig. 13d). We attribute them to phonon modes with different wavevectors, i.e., $k_1$, $k_2$, and $k_3$. The phonon modes exhibit broad frequency band owing to the broad wavevector excitation of CPW and high phonon velocity. The velocity extracted from a linear fit of the dispersion estimated from the three observed modes is found to be ~2.5 km s$^{-1}$ (Fig. 2j), which is about three times larger than the group velocity of magnons derived from mode Y (Fig. 2i). The magnon–phonon coupling within LSMO cannot account for the hybridized mode X in view of the fact that mode X vanishes in the transmission spectra on a bare LSMO film without BFO (see Supplementary Fig. 4). A neutron scattering experiment[32] reported the velocity of longitudinal phonon mode to be 2.6 km s$^{-1}$, which is fairly close to the value (3.2 km s$^{-1}$) estimated from our experiments on a pure BFO sample (see Supplementary Fig. 14). Therefore, we assign mode X to the BFO phonon mode hybridized with LSMO magnon mode. The velocity of mode X (2.5 km s$^{-1}$) is slightly lower than that of phonons in pure BFO[32,47] due to the hybridization with the slow magnon mode. We also note that $k_3 \approx 16$ rad μm$^{-1}$ (Supplementary Fig. 2i). The corresponding wavelength amounts to 0.39 μm which is 20 times smaller than the IDT-driven magnetoacoustic waves in ref. [22].

When the $k_3$ phonon mode (field-independent branch) meets the magnon mode (field-dependent branch), the anticrossing-like feature is observed as shown in Fig. 3a revealing the magnon–phonon hybridization (white arrow). By rotating the in-plane magnetic field of 100 mT, the magnon–phonon hybridized mode forms mode X right above mode Y, as shown in

Supplementary Fig. 13. Essentially, the long decay length of mode X arises from the magnon–phonon hybridization. Figure 3b shows the simulation results on the dispersion relations where magnon and phonon branches are present. In this simulation, we set the hybridization parameter $b_{\mathrm{eff}} = 2 \times 10^5$ (see Supplementary Sec. VI), which corresponds to the anticrossing gap of order 10 MHz (Supplementary Fig. 8). Since the FMR linewidth is around 7 MHz, the magnon–phonon interaction in our setup does not fully enter the strong coupling regime[33,34]. The crossing point is at the wavevector around 20 rad μm$^{-1}$, which is beyond the $k_3$ mode of the CPW excitation (Supplementary Fig. 2i). Therefore, the main features of Fig. 3a may result from magnon–phonon mode hybridization off the crossing point as indicated by the green circle in Fig. 3b. Although here we assume the coupling type to be anticrossing, we admit that it is hard to distinguish anticrossing from level-attraction type[48–50] which may induce signal enhancement[51] and line narrowing[52] around the crossing point, especially with intermediate coupling strength (Supplementary Fig. 10).

The group velocities of the magnon–phonon hybridized mode are independently characterized by the TR-BLS measurements[53] as shown in Fig. 4. The magnon–phonon hybridized mode is excited from the NSL antenna and detected by the focused laser spot of TR-BLS at two different probing positions, i.e., P1 and P2 being 5 μm apart as illustrated in the microscope image (Fig. 4a). To excite the mode X, we adjust the magnetic field to 100 mT at the angle of 54° to the [110] crystal direction of LSMO and set the microwave pulse frequency to 7.5 GHz close to that of mode X shown in Fig. 1d. The TR-BLS measures the magnetic signals on the BFO/LSMO heterostructure (Fig. 4b), where rising and falling edges[54] are observed corresponding to switching on and off of the microwave excitation at the NSL antenna. Due to a propagation distance $d = 5$ μm from P1 to P2, the time-resolved BLS response indicates a certain time delay △$t$ as observed both in the rising edges (Fig. 4c) and falling edges (Fig. 4d). If one extracts △$t$ at 50% amplitude in rising and falling edges as indicated in Fig. 4c, d, an average time delay of about 2 ns is obtained, which yields an average group velocity of about 2.5 km s$^{-1}$. This high velocity is consistent with the velocity extracted from Fig. 2j and therefore is attributable to mode X being the magnon–phonon hybridized mode. The TR-BLS results for a greater separation of 55 μm (Supplementary Fig. 15) exhibit a longer time delay. In Fig. 4b, a weak mode indicated by the blue ellipses both in rising and falling edges might imply the contribution of a slow magnon mode. The average time delay for the slow magnon mode △$t'$ is extracted from Fig. 4b to be around 13 ns, which yields a low group velocity of ~0.4 km s$^{-1}$. This value agrees reasonably well with the angle-dependent group velocities of mode Y estimated from AR-PSWS measurements as shown in Supplementary Fig. 16.

**Millimeter-long magnon decay length induced by magnetoelastic coupling.** The transmission amplitude of mode Y is found to decay exponentially and a fitting yields a magnon decay length $\lambda_{\mathrm{m}} = 5$ μm (red symbols in Fig. 5). On the contrary, the transmission amplitude of mode X decreases much less with $d$. At a propagation distance around 50 μm, the transmission signal strength of mode X can still be observed by TR-BLS (Supplementary Fig. 15) and AR-PSWS (Fig. 5 and Supplementary Fig. 17). In AR-PSWS we detect signals at even $d = 1060$ μm (Supplementary Fig. 18). These results suggest that mode X and mode Y exhibit quite different propagation properties. To quantify the attenuation of the two modes, we fitted the

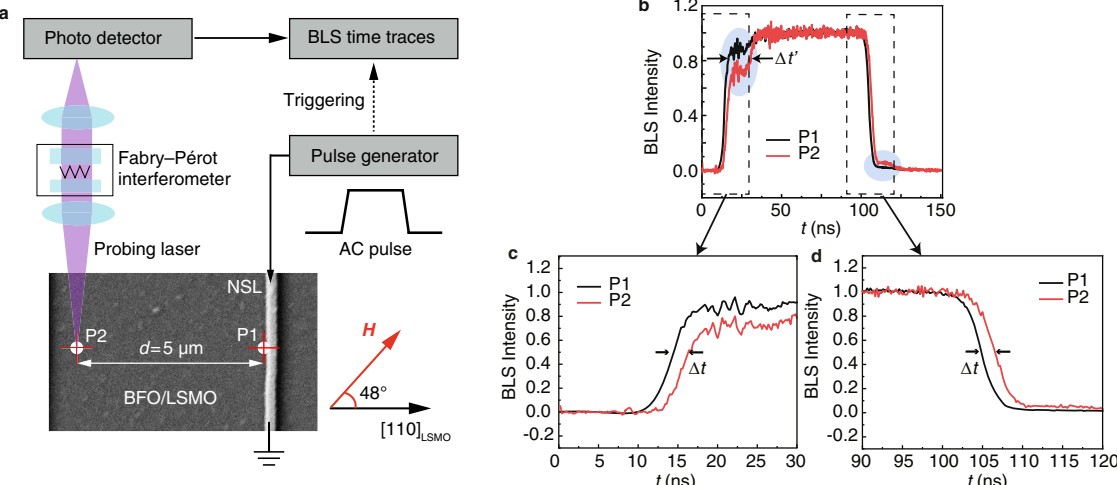

**Fig. 4 Time-resolved BLS measurements on the BFO/LSMO heterostructure. a** Sketch of the time-resolved BLS setup and the SEM image of the investigated sample. Probing positions P1 and P2 are depicted in the SEM image with a separation distance $d = 5\,\mu m$. The measurement is conducted at 100 mT with the external field applied 48° to the [110] crystal direction of LSMO. **b** A microwave pulse with the frequency of 7.5 GHz is injected into the NSL antenna. Rising and falling edges are observed at P1 (black line) and P2 (red line) by time-resolved BLS. The blue ellipses mark the slow magnon mode with a time delay $\Delta t'$. **c** A rising in BLS intensity is first observed in P1 which is close to the NSL antenna, and the rising edge arrives at P2 with a time delay $\Delta t$. **d** After the microwave pulse, the falling of the BLS intensity is first observed in P1 and then arrives at P2 with the time delay $\Delta t$.

transmission data of Fig. 5 by

$$S_{21} = C_{m}\exp\left(-\frac{x}{\lambda_{m}}\right) + C_{mp}\exp\left(-\frac{x}{\lambda_{mp}}\right) \quad (2)$$

where $C_{m}$ and $C_{mp}$ are two distance-independent prefactors, $\lambda_{m}$ and $\lambda_{mp}$ are magnon decay lengths for pure magnon and hybridized magnon–phonon contributions. For the mode Y data points (red circles in Fig. 5), we set $C_{mp} = 0$ and the fitting yields $\lambda_{m} = 5\,\mu m$ as discussed above. In the near-antenna region, mode X experiences a fast decay attributed to the pure magnon contribution as shown in Fig. 5 and the BLS spatial mapping near the NSL (see Supplementary Fig. 19). In the region away from the NSL (35–58 $\mu m$), the pure magnon mode vanishes and the corresponding 2D mapping of BLS exhibits a much slower decay amid weak acquired magnon signals (see Supplementary Fig. 20). The pure magnon decay length $\lambda_{m} = 5\,\mu m$ is used for fitting the mode X data (black squares and triangles in Fig. 5) with $C_{m}, C_{mp}, \lambda_{mp}$ being the fitting parameters. Then it yields $\lambda_{mp} = 1\,mm$, confirming an ultra-long decay length for the hybridized magnon–phonon mode. Note that this decay length is even larger than that for pure magnons in insulating thin-film YIG[10] and is typical of acoustic waves at the GHz frequency range[20] as experimentally demonstrated in a bare BFO sample with integrated IDTs (see Supplementary Fig. 12). Our work now demonstrates such ultra-long decay lengths for magnon signals in a metallic thin film in an all-magnet heterostructure.

Under some regularity conditions on the dispersion relation as a function of wavevector (see Supplementary Fig. 8), the decay length of a plane-wave mode can be described alternatively by $\lambda = 2v_{g}/\Delta\omega$, where $v_{g}$ and $\Delta\omega$ are the group velocity and the relaxation rate of the mode, respectively. For $\lambda_{m}$, the AR-PSWS yielded $v_{g} = 0.8\,km\,s^{-1}$, and combining it with the FMR linewidth measured by PPMS (Supplementary Fig. 3), we derive $\Delta\omega_{m} \approx \gamma\Delta H \approx 460\,MHz$ implies $\lambda_{m} \approx 3\,\mu m$ where $\gamma = 2\pi \times 29\,GHz\,T^{-1}$ is the gyromagnetic ratio, in reasonable agreement with the fitting of the transmission amplitude by Eq. (2). For $\lambda_{mp}$, the TR-BLS gave $v_{g} = 2.5\,km\,s^{-1}$

so that to account for $\lambda_{mp} = 1\,mm$ requires $\Delta\omega_{mp} = 5\,MHz$, almost two orders of magnitude smaller than the FMR relaxation rate. Therefore, the long decay length for the magnon–phonon hybridized mode partially arises from the enhanced group velocity, but there has to be a significantly reduced relaxation rate too. This change in relaxation rate compared to that of pure magnons provides further evidence on the hybridization with phonons. Further experimental and theoretical studies are desired to fully unveil the origin of ultra-low relaxation rate of the magnon–phonon hybridized mode in the BFO/LSMO heterostructures.

**Summary and outlook.** In summary, our work has demonstrated long-distance magnon propagation in BFO/LSMO heterostructures. This is realized by the magnon–phonon coupling at the BFO/LSMO interface. As revealed by the AR-PSWS measurements, the pure magnon mode in LSMO shows a short decay length of ~5 $\mu m$, whereas the magnon–phonon hybridized mode exhibits an ultra-long magnon decay length of up to 1 mm. The group velocity of the magnon–phonon hybridized mode reaches about 2.5 km s$^{-1}$ characterized by TR-BLS, and is more than three times higher than the pure magnon mode in LSMO film. The numerical simulations reveal that the magnetoelastic coupling at the BFO/LSMO interface is responsible for the formation of the magnon–phonon hybridized mode. Our results provide an effective solution to significantly enhance the magnon decay length via magnon–phonon coupling in metallic magnets and thus represent key advancement in hybrid magnonics[23]. The conventional DE spin-wave mode (mode Y) shows clear nonreciprocity, while the magnon-polaron mode (mode X) is rather reciprocal. The different behavior in spin-wave non-reciprocity may be potentially useful for spin-wave computing and signal processing[55]. The reciprocal property of mode X may be further studied in future works. In view of the multiferroic nature of BFO, future investigation may lead to voltage manipulation[56] of the magnon–phonon hybridized mode by switching the ferroelectric polarization.

## Methods

**Sample preparation.** BiFeO$_3$/(La$_{0.67}$Sr$_{0.33}$)MnO$_3$ were deposited on NdGaO$_3$ (001) substrate using KrF excimer pulsed laser deposition (PLD) with energy of ~1.5 J cm$^{-2}$,

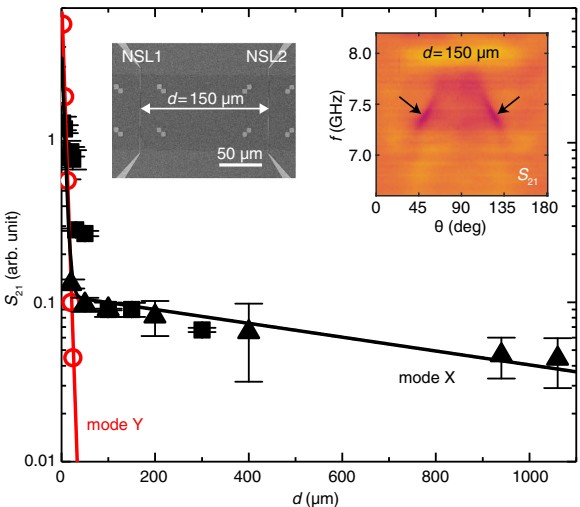

**Fig. 5 Long-distance propagation of magnon–phonon hybridized mode at the BFO/LSMO interface.** Transmission amplitude $S_{21}$ plotted in a logarithmic scale as a function of the distance $d$ between excitation and detection antennas. The left inset shows the SEM image of Sample Z13 with $d = 150\,\mu m$ as an example. The red open circles are data points extrapolated for mode Y from different samples as shown in Fig. 2a–e. The red solid line is a fit considering an exponential magnon decay, which yields a magnon decay length $\lambda$ to be ~5 $\mu m$ for mode Y. The black squares (triangles) are data points taken from samples with NSL (CPW) antennas for mode X (Supplementary Figs. 17 and 18). The black solid line is a fit based on Eq. (2) considering the contributions from a pure magnon mode with a short decay length $\lambda_m \approx 5\,\mu m$ in superposition to a magnon–phonon hybridized mode with a long decay length of $\lambda_{mp} = 1\,mm$. The right inset shows angle-dependent transmission spectra $S_{21}$ measured on Sample Z13 with an applied field of 100 mT and a propagation distance $d = 150\,\mu m$, where the black arrows indicate two branches of mode X attributed to the magnon–phonon hybridized mode.

wavelength of 248 nm and repetition rate of 5 Hz. The $(La_{0.67}Sr_{0.33})MnO_3$ (LSMO) film was first deposited on substrate at 780 °C with oxygen partial pressure of 25 Pa. $BiFeO_3$ (BFO) was then deposited on top of LSMO layer at 700 °C with oxygen partial pressure of 20 Pa. After the deposition, the sample was annealed at $2 \times 10^4$ Pa oxygen partial pressure for 20 min and then cooled down to room temperature with the rate of 10 °C/min.

**Scanning transmission electron microscopy (STEM) and energy-dispersive X-ray spectroscopy (EDS).** The cross-sectional STEM specimens were thinned to less than ~30 μm first by using mechanical polishing and then by performing argon ion milling (PIPSTM Model 695, Gatan Inc.). The annular dark field (ADF)-STEM and EDS images were recorded at 300 kV using an aberration-corrected FEI Titan Themis G2 with the convergence semi-angle for imaging 30 mrad and the collection semi-angles snap 39–200 mrad.

**Scanning-NV magnetometry.** Images of the magnetic stray fields and sample topography have been obtained using a commercial nitrogen-vacancy (NV) scanning probe microscope (ProteusQ, Qnami AG) equipped with a diamond sensing tip hosting a single NV center located in its apex. The NV spin can be initialized optically, manipulated by microwave magnetic fields and read-out by the NV's spin-state-dependent fluorescence (PL). For measurements, the tip is scanned in contact with the sample via frequency modulated atomic force microscopy. Simultaneously to the topographic imaging the nitrogen-vacancy center is spectroscopically probed, which enables to quantitatively infer the local magnetic stray fields emerging from the sample surface on the nanometer scale (the measurement method is described in more detail e.g., in ref. [57]). We additionally utilize 'quenching mode' imaging, where the state-mixing of the NV center is recorded via a drop in its PL (as described e.g., in ref. [58]). This PL reduction depends on the magnitude of the magnetic stray fields perpendicular to the NV quantization axis (transversal fields) and therefore does not determine the specific field direction. Hence, this mode commonly remains qualitative, but provides valuable information on magnetic textures in the sample and is here utilized to measure the regularity and periodicity of the magnetic stray fields in larger areas.

**PhaseFMR characterization with PPMS.** The ferromagnetic resonance (FMR) measurement was carried out in PhaseFMR-40 (Nanosc Instrument) which was integrated in the chamber of Physical Properties Measurement System (PPMS, Quantum Design). The coplanar waveguide is used for the ferromagnetic resonance (CPW-FMR) excitation. During the measurement, the frequency injected into the CPW was fixed and the external magnetic field was swept along the in-plane <110> direction of the LSMO film, then other frequencies were chosen to repeat the measurement. The collected data was fitted by Kittel function and Lorentzian function to extract the gyromagnetic ratio and damping constant.

**Angle-resolved propagating spin-wave spectroscopy (AR-PSWS).** The angle-resolved spin-wave measurement was conducted with the in-plane external magnetic field fixed at 100 mT, while its orientation with respect to the [110] crystalline of the LSMO film was swept within the film plane from −180º to +180º at a step of 2º. At each angle, microwaves of frequencies from 5 to 9 GHz were continuously injected by a vector network analyzer (ROHDE & SCHWARZ ZVA 40) into the antenna 1 (antenna 2) to excite spin waves in the underneath sample, in the meantime, the propagating spin waves was detected by the antenna 2 (antenna 1) on the other side, in such way the spin-wave transmission spectra $S_{21}$ ($S_{12}$) was obtained. The reflection spectra $S_{22}$ ($S_{11}$) was obtained when the antenna 1 (antenna 2) excited spin waves were simultaneously detected by antenna 1 (antenna 2). The power of the injected microwave from the vector network analyzer is −10 dBm. All the angle-resolved spin-wave measurements were undertaken at room temperature.

**Time-resolved Brillouin light scattering (TR-BLS).** The time-resolved Brillouin light scattering measurement was conducted with 100 mT external field. A microwave pulse with the frequency of 7.5 GHz was applied with a 100 ns duration, 3.19 MHz repetition rate, and power of 22 dBm. All the measurements were undertaken at room temperature. The data presented in Fig. 3 are measured with a blue laser (473 nm) with the external field applied 48° to the [110] crystallization of LSMO. The data presented in Supplementary Fig. 15 are measured with a green laser (532 nm) with the external field applied 50° to the [110] crystallization of LSMO.

**Reporting summary.** Further information on research design is available in the Nature Research Reporting Summary linked to this article.

## Data availability
The authors declare that the main data supporting the findings of this study are available within the article and its Supplementary Information files. Extra data are available from the corresponding authors upon reasonable request.

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

## Acknowledgements
The authors thank Y. Zhang, E. Guo, H. Chen, J. Xiao, J.-X. Zhang, G. Bauer, X.-R. Wang, O. Gomonay, K. Harii, and J.'i Ieda for helpful discussions. The authors acknowledge support from the National Natural Science Foundation of China (Grant No. 12074026 and U1801661), 111 talent program B16001, the National Key Research and Development Program of China Grants No. 2016YFA0300802 and 2017YFA0206200, and Shenzhen Institute for Quantum Science and Engineering, Southern University of Science and Technology (Grant No. SIQSE202007). The work in Tsinghua University is supported by the Basic Science Center Program of NSFC (grant no. 51788104). Y.S. and P.G. acknowledge the support by Key-Area Research and Development Program of Guangdong Province (2018B030327001, 2018B010109009) and Electron Microscopy Laboratory of Peking University for the use of electron microscopes. K.Y. is supported by JST, PRESTO Grant Number JPMJPR20LB, Japan, and JSPS KAKENHI (No. 19K21040 and No. 21K13886). S.M. is supported by JST CREST Grant (No. JPMJCR19J4, No. JPMJCR1874, and No. JPMJCR20C1) and JSPS KAKENHI (Nos. 17H02927 and 20H01865) from MEXT, Japan. The numerical simulations were conducted using the supercomputer HPE SGI8600 in the Japan Atomic Energy Agency. M.H. and D.G. thank SNSF for financial support via grant 177550.

## Author contributions
J.Z., J.C., and H.Y. conceived and designed the experiments. M.C., J.M., and C.-W.N. provided the BFO/LSMO films and characterized the samples by PhaseFMR techniques with PPMS. H.W., J.C., D.Y., J.-Ph.A., and H.Y. designed and fabricated the spin-wave devices. K.W. and P.M. conducted the NV magnetometry measurements. Y.S., P.G., and D.Y. conduct the TEM characterization. J.Z., J.C., H.W., and H.Y. performed the AR-PSWS measurements. M.H. and D.G. conducted the TR-BLS experiments. J.Z., J.C., H.W., M.H., D.G., and H.Y. analyzed the experimental data. K.Y. and S.M. performed the theoretical modeling and simulations. D.G., C.-W.N., and H.Y. supervised the experimental study. H.Y., J.Z., J.C., H.W., K.Y., C.-W.N., J.-Ph.A., S.M., and D.G. wrote the paper and the supplementary information.

## Competing interests
The authors declare no competing interests.
