## [Peer review file · Nature Communications]

REVIEWER COMMENTS

Reviewer #1 (Remarks to the Author):

In the manuscript, Zhang et al., present the experimental results on a magnon decay length of up to one millimeter in multiferroic/ferromagnetic BiFeO₃(BFO)/La_{0.67}Sr_{0.33}MnO₃(LSMO) heterostructures using angle-resolved propagating spin-wave spectroscopy and time-resolved Brillouin light scattering. The authors reveal the nature of ultra-long magnon decay length based on the magnon-phonon coupling at the interface of BFO/LSMO. The high group velocity extracted from the experiments and long relaxation time assumed in the manuscript are required to attain 1-mm decay length. I find these experimental results are interesting and could have an impact on hybrid magnonics, which is rapidly growing. However, in my opinion, it is not convincing that the long decay length of magnons in BFO/LSMO arises from the magnon-phonon coupling and therefore, more experimental and/or theoretical data are needed to make a solid conclusion.

(1) The decay length of an acoustic wave at GHz frequency is typically in the millimeter range, as written in the manuscript. In a hybrid system, e.g., Refs. 14 and 29, magnons can benefit from the hybridization with long-lifetime phonons, resulting in long-distance propagation. In the manuscript, I didn't find the authors show the experimental evidence for long-lifetime acoustic phonons in BFO. Instead, they mentioned some references (e.g., Refs. 24 and 25) about the phonon lifetime. From these references, I cannot conclude the acoustic phonons of 2-6 GHz in BFO host very long lifetime that allows for a propagation distance of 1 mm. The authors should add more relevant data to address this point.

(2) Let's assume that the acoustic phonons with a long lifetime in BFO strongly couple to the magnons in LSMO. According to the formula of $\lambda = 2vg/\Delta\omega$ written in the manuscript, the minimum value of $\Delta\omega$ for a hybridized mode can be simply estimated by $(\Delta\omega_{\text{magnon}} + \Delta\omega_{\text{phonon}})/2$, which is dominated by $\Delta\omega_{\text{magnon}}$. With considering enhanced group velocity vg , the magnon decay length in BFO/LSMO will increase by one order of magnitude, rather than > two orders of magnitude stated in the manuscript. Can the authors comment on this argument?

(3) In Fig. S11d, the authors attribute three absorption dips to the acoustic phonons. From the data, I find the phonon linewidth, related to $\Delta\omega_{\text{phonon}}$, is quite large. It seems that the phonon lifetime is rather short, which conflicts with the statement in the manuscript. Can the authors comment on this? In addition, please describe the excitation mechanism of phonons using CPWs.

(4) The 1-mm magnon decay length was extracted from Fig. S15. The signal is weak for $d > 100 \mu\text{m}$. I see a bit stronger signal at a magnetic field $> 100 \text{ mT}$, especially for $d = 940$ and $1060 \mu\text{m}$. The spin-wave frequency is larger than 6 GHz for the field strength $> 100 \text{ mT}$. In Fig. S11d, the phonon frequency is below 6 GHz and is magnetic field independent. How does the magnon-phonon coupling occur for a magnetic field $> 100 \text{ mT}$?

(5) The authors analyzed magnon-phonon coupling based on a numerical model. They assume the coupling only exists at the interface. What is the thickness of the interface ($t_{\text{interface}}$)? The interfacial coupling may lead to a difference in the S12 and S12 spectra. The effect might be small since the measurement geometry is away from the Damon-Eshbach configuration. Can the authors compare S21 to S12 measured in BFO/LSMO?

(6) The intensity of the mode X calculated by the numerical model (Fig. 2f-h) doesn't rely on the field angle. However, the mode X should vanish near the field angle of 0 and 90 degree since the coupling coefficient (equation S2) is nearly zero. Can the authors comment on this?

(7) In Fig. S8, the dispersion relations of the phonon and magnon mode are calculated for $\alpha = 0$. In this case, strong magnon-phonon coupling occurs for $\theta \approx 45$ degree and the mode profile

exchanges after the crossing point. However, for $\theta = 0$ and 90° , the magnon-phonon coupling should be weak. I expect no exchange between the phonon and magnon mode. In addition, if the authors take the experimental value of α , the anti-crossing gap is still clear. If yes, I suggest the authors include this part in the main manuscript and discuss the magnon-phonon coupling strength.

(8) The authors mention that the phase-insensitive behavior of the mode X may be related to the coupling with phonons. I am not sure the authors clearly explain this in the text.

(9) Did the authors measure a 2D spatial map of the mode X using micro-BLS? If yes, it is better to include the data in the manuscript.

(10) The S21 spectrum of the mode Y shows much better oscillation behavior in BFO/LSMO (Fig. 1d) than that in pure LSMO (Fig. S4). Is there a physical reason behind?

(11) The introduction does not cover sufficiently the field of magnon-phonon coupling and hybrid magnonics. Since the manuscript is going to demonstrate strong magnon-phonon coupling, I suggest the authors introduce more information on these topics.

Reviewer #2 (Remarks to the Author):

Summary of the work:

In their work entitled Long magnon decay length in BiFeO₃/La_{0.67}Sr_{0.33}MnO₃ heterostructures by J. Zhang et al., the authors report on the experimental observation of unprecedented long spin wave decay lengths up to 1 mm in a heterostructure comprised of a multiferroic BiFeO₃ and a La_{0.67}Sr_{0.33}MnO₃ layer underneath which is typically in the half-metallic ferromagnetic phase for this composition. The authors attribute this ultra-long decay length to the formation of a hybrid magnon-phonon mode (denoted X mode from the authors) which propagates with a much higher group velocity than the "normal" spin wave mode and an associated ultra-low relaxation rate of this hybridized state. The authors used a set of different experimental methods such as angular-dependent (the angle between the wave-vector and the in-plane externally applied static magnetic field) propagative spin wave spectroscopy using different devices where they varied the distance between the two antennae to show the spin wave transmission about long distances. The authors used them as well as for the determination of the group velocity. These findings were confirmed by performing time-resolved Brillouin Light Spectroscopy as well. Following a theoretical model from Yamamoto et al., the authors explain their findings by a simplified model where- via an isotropic magnetoelastic coupling- the spin wave modes in the LSMO film couple with a longitudinal acoustic wave mode. The experimental findings are in a good agreement with the simulations performed based on this model, correctly reproducing the most important findings.

Comments on validity, clarity etc.

The experimental data and the associated theoretical model are well presented, and the utilized methods are all described as well. In my view, the overall data analysis and interpretation is sound and well justified. The article's findings is original and the topic is timely and contributes to the interdisciplinary activities in the utilization of different hybridized systems, enriching various research areas beyond pure magnonics. The article is written in a clear and concise manner. The supplementary material is detailed and further elucidates the main statements of the paper.

However, although there is a good agreement with the experiment and the theory based on magnon-phonon hybridization, partially the proof of the existence of such a hybrid state might be not fully clear for the non-expert reader in the main text. Thus, I have some questions/remarks to the authors.

Questions and remarks to the authors

Naively, when speaking of a hybridized state between two distinct systems with different dispersions, I expect to see some sort of avoided level crossing spectra, even if the system is not fully strongly

coupled, which would still be reflected in the corresponding reflection/transmission spectra. The authors devote the supplementary material section VI. to show calculated, not experimental, dispersion relations only in the supplementary material and in one figure in section IX. (Fig S11) of the same supplementary material. However, I cannot see that type of feature in Fig. S11 shown for specific k . Possibly a zoom to the anticrossing region might be helpful and answer already my concern. Additionally, as showing the existence of such hybrid mode is crucial for the statements of this work, I personally think it might be elucidating if this figure appears in the main text.

Regarding that, the LA velocity of the mode in BFO is reported to be 2.5 km/s which is the same value as the one found for the X-mode. Hence, is there really a propagation of the once, by the first antenna, excited hybrid mode or do we have here the coupling between a propagating phonon which- by the magnetoelastic coupling- is "exciting magnons along the way", such that it is seen by the utilized experimental methods probing the magnons? Could the authors please comment on that? Further, in the main text the authors write when discussing the hybridized state in line 177-178 "enhancement in the transmission spectra is observed" but in the supplementary material they also write in section VI. "[...] reflected upon the strong enhancement of the S22 signal which is not related to the magnon-phonon hybridization. " Could the authors please clarify?

It is known from cavity magnonics that the linewidths are changing towards the hybridization. Today, one distinguishes between coherent coupling with anticrossings as their hallmark and dissipative couplings with level crossings. It was shown that in the presence of dissipative coupling the linewidth of the hybrid state is decreased, and the amplitude increases. Furthermore, similar to some works in cavity magnonics (One tone approach from Harder et. al, PRL 2018 two tone approach, theory: Grigoryan et al. PRB 2018, experiment: Boventer et al., PRR 2020), the coupling strength becomes complex or purely imaginary as is the expression of the coupling strength from the authors in the supplementary material.

I wonder if the authors have considered the possibility of such type of hybridization while seeing 'only' an enhancement in the spectra at the position of the conditions for the hybridization (Fig. S11) and if it could at least partially account for the observation of such ultra-low relaxation rate the authors reported. Could they please also comment on that?

What would be the lowest coupling strength possible to still see such long decay length?

In the following I only have minor comments.

- a) What is the initial direction of the, for instance, out-of plane component, of the ferroelectric polarization state of the BFO layer? How does it affect the hybrid X mode if that state would be flipped, as it has been shown that the ferroelectric domain polarization state also can have an impact?
- b) The scale bars are only mentioned in the captions. I would suggest adding the scales rather to the image, helping to grasp the entire figure even faster.
- c) What is the orientation of the NGO substrate, 001 or 110, for instance?
- d) Line 258: The value of 29 GHz/T for the gyromagnetic ratio is a fit result and hence not 28 GHz/T?

Recommendation:

In conclusion the observation of such a long decay length marks a significant advance for magnonics and beyond. The utilization of such hybridized mode opens new routes for the integration of magnonic devices to other architectures and opens new research directions based on such BFO/LSMO heterostructures as it can merge the functionalities of hybridized states and functional oxides together.

Thus, in principle, I can recommend this work for publication after minor revision in Nature Communications if the nature and its experimental demonstration of the hybridized state is further clarified.

Yours sincerely,
Reviewer

Reviewer #3 (Remarks to the Author):

The paper presents experimental evidences of waves propagating in BFO/LSMO bilayer deposited on a NGO substrate. Two modes are evidenced. The slower one is the dipole exchange mode propagating perpendicularly to the magnetization direction in LSMO. The faster one propagates with a group velocity of 2.5 km/s. This latter one is interpreted as an acoustic wave propagating in BFO. The experimental results are nice but the interpretation is questionable (see the attached file). I suggest major revisions before publication.

The paper presents experimental evidences of waves propagating in BFO/LSMO bilayer deposited on a NGO substrate. Two modes are evidenced. The slower one is the dipole exchange mode propagating perpendicularly to the magnetization direction in LSMO. The faster one propagates with a group velocity of 2.5 km/s. This latter one is interpreted as an acoustic wave propagating in BFO. The experimental results are nice but the interpretation is questionable because 2.5 km/s corresponds to shear wave velocity in LSMO [Phys. stat. sol. (a) 195, No. 2, 350–358 (2003)]. According to [Appl. Phys. Lett. 102, 182905 (2013)] the BFO shear elastic constant C_{66} is 89 GPa. Taking a BFO mass density of 8220 kg/m³, we obtain a shear wave velocity of 3.3 km/s. According to [Eur. Phys. J. B 94, 108 (2021)] the NGO shear modulus is 90 GPa. Taking a NGO mass density of 7.3 kg/m³, we obtain a shear wave velocity of 3.5 km/s. According to these estimations, the LSMO shear wave velocity is lower than those in BFO and NGO. This means that LSMO sandwiched between BFO and NGO is an acoustic wave guide. Consequently the observed wave propagating at 2.5 km/s is localized within LSMO. Coupling acoustic waves and dipole exchange waves within LSMO is more likely than coupling waves in different media BFO and LSMO. Concerning the model, could the authors explain why the exchange bias field has the same in-plane and out-of-plane contributions and why the in-plane contribution depends not on magnetization direction?

Response to referee reports

First of all, we thank all three reviewers for their time and efforts and for their in-depth comments and suggestions which we find very constructive and helpful. We took them into account fully and carefully revised the manuscript which we believe has significantly improved. We are therefore optimistic that the manuscript now meets the requirements for publication in *Nature Communications*. Below we provide our point-to-point responses to all of the reviewer comments.

Reviewer #1:

In the manuscript, Zhang et al., present the experimental results on a magnon decay length of up to one millimeter in multiferroic/ferromagnetic BiFeO₃(BFO)/La_{0.67}Sr_{0.33}MnO₃(LSMO) heterostructures using angle-resolved propagating spin-wave spectroscopy and time-resolved Brillouin light scattering. The authors reveal the nature of ultra-long magnon decay length based on the magnon-phonon coupling at the interface of BFO/LSMO. The high group velocity extracted from the experiments and long relaxation time assumed in the manuscript are required to attain 1-mm decay length. I find these experimental results are interesting and could have an impact on hybrid magnonics, which is rapidly growing. However, in my opinion, it is not convincing that the long decay length of magnons in BFO/LSMO arises from the magnon-phonon coupling and therefore, more experimental and/or theoretical data are needed to make a solid conclusion.

Response:

We thank Reviewer #1 for finding our work “**interesting and could have an impact on hybrid magnonics**”. As mentioned, we have done additional experimental and theoretical work to further consolidate this conclusion, as detailed in the following.

(1) The decay length of an acoustic wave at GHz frequency is typically in the millimeter range, as written in the manuscript. In a hybrid system, e.g., Refs. 14 and 29, magnons can benefit from the hybridization with long-lifetime phonons, resulting in long-distance propagation. In the manuscript, I didn't find the authors show the experimental evidence for long-lifetime acoustic phonons in BFO. Instead, they mentioned some references (e.g., Refs. 24 and 25) about the phonon lifetime. From these references, I cannot conclude the acoustic phonons of 2-6 GHz in BFO host very long lifetime that allows for a propagation distance of 1 mm. The authors should add more relevant data to address this point.

Response:

In order to address this point, we have prepared a new sample of pure BFO and integrated two interdigital transducers (IDTs) to excite and detect phonons over different distances being 50 μm and 1 mm, respectively.

One can see from the transmission spectra S_{21} that the signal strength for the $d=1$ mm sample is only slightly weaker than that of the $d=50$ μm sample, which suggests that the decay length of phonons in BFO is approximately 1.7 mm with a rough estimate. The phonon velocity in pure BFO is also estimated to be 3.2 km s^{-1} , which is slightly higher than 2.6 km/s quoted in Ref. [32] and 2.5 km/s in our main data. The difference may be attributed to the effect of substrate, different crystalline orientations, and the hybridization with magnons, as discussed below in response to Reviewer #3.

We have now included these additional experimental data in the supplementary information as Section X, and added a sentence in the main text “...as experimentally demonstrated in a bare BFO sample with integrated IDTs (see Supplementary Fig. 14).”

(2) Let’s assume that the acoustic phonons with a long lifetime in BFO strongly couple to the magnons in LSMO. According to the formula of $\lambda = 2vg/\Delta w$ written in the manuscript, the minimum value of Δw for a hybridized mode can be simply estimated by $(\Delta w_{\text{magnon}} + \Delta w_{\text{phonon}})/2$, which is dominated by Δw_{magnon} . With considering enhanced group velocity vg , the magnon decay length in BFO/LSMO will increase by one order of magnitude, rather than $>$ two orders of magnitude stated in the manuscript. Can the authors comment on this argument?

Response:

We thank Reviewer #1 for this very insightful discussion. In a very ideal (theory/simulation) scenario where all parameters (i.e. the frequency f , the wavevector k and also the magnetic field H) are finely tuned to reach a crossing point, the anti-crossing (or level-repulsion) should occur, which could be observed with a fixed H and varying k , and alternatively with a fixed k and varying H as shown in the simplified simulation results considering the coupling of magnon and phonon branches. Indeed, in either case, at the crossing point, the linewidth should be only slightly narrower than that of pure magnons.

One trivial issue for simulation but non-trivial issue for experiments, is how to efficiently excite magnon and/or phonon with high wavevector k of about 20 $\text{rad}/\mu\text{m}$. In all our experiments, we have to admit that we did not manage to efficiently excite modes with wavevectors of up to 20 $\text{rad}/\mu\text{m}$ as one may see from the wavevector profile of the CPW calculated from Fourier transformation as shown in the red dashed line in the newly added Fig. 3b (we copy it here for the convenience) and likewise for nanostriplines (Figs. S2g and h). However, the high-order wavevector k_3 (green circle) is located near the crossing point and the excitation can no longer be considered as 100% pure phonons, but instead contains a magnetic (magnon) information. In other words, the mode X observed experimentally may well be the “green circle” mode being a hybridized mode off the crossing point, which contains magnon information and yet, almost fully retains its phonon high velocity and long life-time.

The key for the observation of the “green circle” mode is the great sensitivity of the piezoelectric effect, i.e. $D = d * T$ where D is the electric flux density, T is the stress, and d is the piezoelectric coefficient, which is very large indeed. Thus, a small stress generates a large voltage. Instead, even a large current can cause only a minor deformation to the crystal. This has two consequences:

1. On the excitation antenna, the rf current mainly drives magnetization dynamics while the direct excitation of acoustic waves is very small. The rf current drives non-resonant magnons and the magnons generates an effective force field on the phonons owing to the magneto-elastic coupling. The phonon amplitude can be estimated as $A_{\text{phonon}} \approx \frac{g_{\text{eff}}}{\Delta\omega} A_{\text{magnon}}$, where g_{eff} is the coupling strength and $\Delta\omega$ is the difference in the resonance frequencies. Even when the phonon and magnon are separated by 1GHz, there is still a 5% hybridization between them. The main part of the phonon signal in our experiments comes from this magnon-phonon hybridization, not from the inverse piezoelectric excitation by the rf current.
2. On the receiving antenna, in particular at large distances, the signal is dominated by the direct piezoelectric effect. This is consistent with the fact that the long-distance signals do not show phase oscillation (the signal is always entirely negative). This antenna picks out the phonon-dominated signal which is weakly hybridized with magnons. Still the signal is distinctively magnetic in that it depends strongly on the magnetic field direction.

In summary, the mode X observed in experiments, including the magnetic field dependence and long decay length, can be attributed to a mildly hybridized mode (slightly away from the crossing point) corresponding to the magnon-phonon coupling, as indicated by the green circle in the newly added Fig. 3b showing the dispersion relations.

(3) In Fig. S11d, the authors attribute three absorption dips to the acoustic phonons. From the data, I find the phonon linewidth, related to $\Delta\omega_{\text{phonon}}$, is quite large. It seems that the phonon lifetime is rather short, which conflicts with the statement in the manuscript. Can the authors comment on this? In addition, please describe the excitation mechanism of phonons using CPWs.

Response:

We thank Reviewer#1 for his/her careful observation on the linewidth of phonon mode in supplementary Fig. S11d (now Fig. S13d).

In these data, the broad linewidth of the phonon signal does not arise from the intrinsic damping of phonons, but is rather dominated by the $\Delta\omega$ induced by broad wavevector excitation Δk of the CPW via $\Delta\omega = v_g * \Delta k$. Since phonons have relatively large group velocity v_g , a finite Δk (determined by the width and imperfections of the CPW antennas) will produce a large $\Delta\omega$, much larger than the linewidth of intrinsic phonon lifetime.

To have a rough estimate of this inhomogeneous broadening, we read Δk from Fig. S2i to be roughly about 2 rad/ μm . Taking the phonon velocity of roughly 3 km/s, this would yield a linewidth broadening $\Delta\omega=v_g*\Delta k$ to be about 1 GHz. To clarify this point in the manuscript, we have added one sentence in the main text: “*The phonon modes exhibit broad frequency band owing to the broad wavevector excitation and high phonon velocity.*”

We agree with the reviewer that the CPWs are designed for the purpose of exciting and detecting magnon signals. Therefore, in most cases, we do not observe clear phonon signals. Nevertheless, since CPWs are directly deposited on top of the multiferroic BFO, in some cases the CPWs may generate oscillating electric field and act as incompetent (or shorted) interdigit transducers (IDTs) that may trigger some phonon modes in BFO via piezoelectric effects.

(4) The 1-mm magnon decay length was extracted from Fig. S15. The signal is weak for $d > 100 \mu\text{m}$. I see a bit stronger signal at a magnetic field $> 100 \text{ mT}$, especially for $d = 940$ and $1060 \mu\text{m}$. The spin-wave frequency is larger than 6 GHz for the field strength $> 100 \text{ mT}$. In Fig. S11d, the phonon frequency is below 6 GHz and is magnetic field independent. How does the magnon-phonon coupling occur for a magnetic field $> 100 \text{ mT}$?

Response:

We thank the reviewer for pointing out this observation. Below we put the data in supplementary Fig. S11d (now Fig. S13d) with $d=12 \mu\text{m}$ as (a) together with those for $d=940 \mu\text{m}$ as (d) for comparison. We can see that the enhancement around 7 GHz, clearly visible in (a) still exists in (d). So this main feature of the data is consistent at all distances. We agree with the reviewer that there does exist some sizable signal above 100 mT and around 10 GHz. This additionally observed mode may be due to even higher order excitation of the CPW antenna. In fact, we checked the two meander CPWs patterned on these two samples (b and e) by SEM and found the widths of the antenna signal and ground lines to be slightly different, due to the lithography process. Such mild differences may have little impact on the k_1 mode, but may impact significantly the high order excitations, such as k_3 and k_4 as shown in the Fourier transformation results, as shown in (c) and (f). The signal above 100 mT around 10 GHz may be associated with the k_4 excitation of the CPWs due to a slight change in the width of the conducting lines.

Furthermore, the Fourier transform of the antenna geometry is just a crude approximation to the real wavenumber excitation spectrum. Phonons and magnons are driven by the electric and magnetic fields respectively generated by the rf current in the CPW. And the field profiles should show some extra spatial dependence in the electric field arising from the charge distribution inside the antenna and in the magnetic field owing to the $1/r$ dependence of Oersted field. As explained in our response to point (2) of this referee, the main driving force is magnetic in this experiment. Indeed, the old Fig. S11d showed that the pure phonon excitation is cut-off above 7GHz whereas the magnetic signal is clearly visible up to 12 GHz in old Fig. S11d too. Consistent with our discussion of point (2) above, at these frequencies far away from the crossing point, the magnetization does not excite significant phonons via magneto-elastic effect.

We find these observations in comparing data sets useful and therefore added one sentence in the supplementary information to comment on this as:

“We note that the real CPW excitation spectrum is further broadened from Fig. S2i due to imperfections in fabrication, and spatially inhomogeneous profiles of the induced electromagnetic fields. Thus, the transmission signal above 100 mT and around 10 GHz may be associated with higher order excitation of the antennas.”

(5) The authors analyzed magnon-phonon coupling based on a numerical model. They assume the coupling only exists at the interface. What is the thickness of the interface ($t_{\text{interface}}$)? The interfacial coupling may lead to a difference in the S12 and S21 spectra. The effect might be small since the measurement geometry is away from the Damon-Eshbach configuration. Can the authors compare S21 to S12 measured in BFO/LSMO?

Response:

In the theoretical model in this work, $t_{\text{interface}}$ appears only through the combination shown in Eq. (S4). Since this is also the only place where the “bare” magneto-elastic coupling constant b appears, and since both $t_{\text{interface}}$ and b are fitting parameters of the model, one can estimate only b_{eff} from the comparison with the experimental data, but not $t_{\text{interface}}$ and b individually. The numerical results on Figs. 2f-h and Fig. S7 were produced for $b_{\text{eff}} = 2 \cdot 10^5$. It is impossible to even guess the real value of b , but if we take $b = 2 \cdot 10^6$, we would get $t_{\text{interface}} = 4$ nm. The smaller b is, the larger is $t_{\text{interface}}$.

The question of asymmetry between S12 and S21 is interesting and important, but out of scope of the present study that focuses on the magnon propagation distance. The theoretical model is too simplified for the asymmetric magneto-elastic coupling between the top and bottom surfaces to be incorporated. Specifically, we approximate the spin wave dynamics by averaging its spatial profile over the thickness direction so that there is no distinction between the Damon-Eshbach modes localized on the top and the bottom surfaces.

In the experimental data, the mode Y is dominated by Damon-Eshbach contributions and they clearly show some asymmetry between S12 and S21, as shown in the figure below.

Therefore, the difference exists, but it does not seem to affect the mode X significantly, maybe because they contain less of the Damon-Eshbach contribution as suggested by the reviewer. We would like to leave it for some future studies.

(6) The intensity of the mode X calculated by the numerical model (Fig. 2f-h) doesn't rely on the field angle. However, the mode X should vanish near the field angle of 0 and 90 degree since the coupling coefficient (equation S2) is nearly zero. Can the authors comment on this?

Response:

As the reviewer pointed out, the magneto-elastic coupling coefficient is zero for 0 and 90 degree, and the numerical calculation does reflect this correctly. In the original manuscript, we plotted Figs. 2f-h in the angular range between 30 and 150 degrees so that the vanishing of mode X at 0 is not shown, but it is shown to disappear at 90 degrees. In Fig. S7, the mode X is shown to vanish at 0 and 90 degrees. In the revised manuscript, we have changed the angular range of Figs. 2a-h to 0-180 in order to avoid the confusion. The updated Fig. 2 is shown below.

(7) In Fig. S8, the dispersion relations of the phonon and magnon mode are calculated for $\alpha = 0$. In this case, strong magnon-phonon coupling occurs for $\theta \approx 45$ degree and the mode profile exchanges after the crossing point. However, for $\theta = 0$ and 90 degree, the magnon-phonon coupling should be weak. I expect no exchange between the phonon and magnon mode. In addition, if the authors take the experimental value of α , the anti-crossing gap is still clear. If yes, I suggest the authors include this part in the main manuscript and discuss the magnon-phonon coupling strength.

Response:

We agree with the reviewer's comment and there is no anti-crossing at 0 and 90 degrees, as can be seen in the insets of Fig. S8. The change of color from blue to orange does not indicate an exchange of

magnon and phonon mode, especially at zero anti-crossing at 0 and 90 degrees. Even at the anti-crossing, it is hard to fully distinguish the magnon and phonon mode, since the anti-crossing gap is small compared to the experimental k resolution. To avoid this confusion, we have now made a new Fig. S8 in which the color does indicate the magnon-phonon ratio.

In the simulation, we set a realistic value for the hybridization parameter of $b_{\text{eff}} = 2 \times 10^5$, which does not ensure strong coupling considering the experimental value of alpha as suggested by the reviewer. If we further increase the b_{eff} , it can certainly enter the strong coupling regime as shown in Supplementary Sec. VI. We have now included additional simulation results in the new Fig. 3b as well as Supplementary Sec. VI and added a few sentences in the main text to discuss on this as below.

“Figure 3b shows the simulation results on the dispersion relations where magnon and phonon branches are present. In this simulation, we set the hybridization parameter $b_{\text{eff}} = 2 \times 10^5$ (see Supplementary Sec. VI), which does not fully enter the strong coupling regime [34].

[34] Berk, C. et al. Strongly coupled magnon-phonon dynamics in a single nanomagnet. *Nat. Commun.* **10**, 2652 (2019).

(8) The authors mention that the phase-insensitive behavior of the mode X may be related to the coupling with phonons. I am not sure the authors clearly explain this in the text.

Response:

We thank the reviewer to point this issue out. Using CPWs to excite and detect spin waves based on PSWS technique can routinely give phase information of propagating spin waves (as those in mode Y). However, it seems to us that using similar technique with IDTs to study propagating acoustic waves does not necessarily provide the same clear phase oscillations in the transmission signals. In previous studies (Refs. [15] and [20]) on propagating acoustic waves in LiNbO_3 , one does not observe clear phase oscillations as we do for propagating spin waves.

“This phase-insensitive behavior might be due to the coupling with phonons since the PSWS technique seems not to detect phonon phase information even with IDTs [15,20].”

[15] Xu, M. et al. Nonreciprocal surface acoustic wave propagation via magneto-rotation coupling. *Sci. Adv.* **6**, eabb1724 (2020).

[20] Küß, M. et al. Nonreciprocal Dzyaloshinskii-Moriya magnetoacoustic waves. Phys. Rev. Lett. 125, 217203 (2020).

(9) Did the authors measure a 2D spatial map of the mode X using micro-BLS? If yes, it is better to include the data in the manuscript.

Response:

We thank the reviewer for this great suggestion. We have now added the 2D spatial map of the mode X measured by micro-BLS as Supplementary Fig. 19. The 270 nm-wide NSL antenna (yellow bar) is located at $Y=0$. The external magnetic field is set at 100 mT with $\theta = 48^\circ$ and the frequency is set at 7.5 GHz so that the spatial distribution of mode X is selectively monitored. Due to the long data acquisition time of the BLS setup, the spatial map is taken within a $6 \mu\text{m} \times 20 \mu\text{m}$ square region. One could observe the decay of mode X in the 2D spatial map, particularly in the first few micrometers, where the decay is primarily determined by the magnon decay length.

In the main text, we have now inserted one sentence to refer to these BLS spatial mapping data as

“In the near-antenna region, mode X experiences a fast decay attributed to the pure magnon contribution as shown in Fig. 5 and the BLS spatial mapping near the NSL (see Supplementary Fig. 19).”

(10) The S21 spectrum of the mode Y shows much better oscillation behavior in BFO/LSMO (Fig. 1d) than that in pure LSMO (Fig. S4). Is there a physical reason behind?

Response:

We thank the reviewer for pointing out that the magnon signal in the LSMO sample without BFO is relatively weak particularly in transmission spectra. We attribute this to the difference in the quality of samples and integrated antennas rather than to physical reasons. This sample was produced at the start of this study. In view of this discrepancy, we have grown a new pure LSMO thin film with newly fabricated NSLs on top and replaced the previous results in Fig. S4 with the new measurement data as shown below.

The data on the new pure LSMO sample shows qualitatively the same behaviour except for a much stronger oscillation signal. The distance between two NSLs is $2\ \mu\text{m}$ and the applied field is fixed at 50 mT and rotated from 0 to 180 degree.

(11) The introduction does not cover sufficiently the field of magnon-phonon coupling and hybrid magnonics. Since the manuscript is going to demonstrate strong magnon-phonon coupling, I suggest the authors introduce more information on these topics.

Response:

We have followed the constructive suggestion of the reviewer to reorganize the introduction and inserted several sentences to better introduce the background information on magnon-phonon coupling and hybrid magnonics. The introduction now includes the following:

“Ferromagnetic resonance can be driven by surface acoustic waves (SAWs) via magnon-phonon coupling in a thin magnetic film on a piezoelectric substrate [13]. In the coupling process, phonon may carry spin information [14]. Nonreciprocal propagation of SAWs [15-20] is generated by the magnon-phonon coupling. Long-range transfer of angular momentum between two YIG layers separated by a thick substrate [21] is realized by the magnon-phonon coupling [22]. Direct imaging and quantification of both standing and propagating acoustomagnetic waves was presented in hybrid structures consisting of a piezoelectric substrate underneath Ni [23]. This way, magnon-phonon coupling and excitation by a large-area interdigital transducer (IDT) allowed to transport spin information over a macroscopic distance at specific IDT frequencies [23]. For future hybrid magnonics[24]...”

[13] Weiler, M. et al. Elastically driven ferromagnetic resonance in nickel thin films. *Phys. Rev. Lett.* **106**, 117601 (2011).

[14] Holanda, J., Maior, D. S., Azevedo, A. & Rezende, S. M. Detecting the phonon spin in magnon-phonon conversion experiments. *Nat. Phys.* **14**, 500-506 (2018).

[15] Xu, M. et al. Nonreciprocal surface acoustic wave propagation via magneto-rotation coupling. *Sci. Adv.* **6**, eabb1724 (2020).

[16] Sasaki, R., Nii, Y. & Onose, Y. Magnetization control by angular momentum transfer from surface acoustic wave to ferromagnetic spin moments. *Nat. Commun.* **12**, 2599 (2021).

[17] Yamamoto, K., Yu, W., Yu, T., Puebla, J., Xu, M., Maekawa, S. & Bauer, G. Non-reciprocal pumping of surface acoustic waves by spin wave resonance. *J. Phys. Soc. Jpn.* **89**, 113702 (2020).

[18] Maekawa, S. & Tachiki, M. Surface acoustic attenuation due to surface spin wave in ferro- and antiferromagnets. *AIP Conf. Proc.* **29**, 542 (1976).

[19] Matsuo, M., Ieda, J., Harii, K., Saitoh, E. & Maekawa, S. Mechanical generation of spin current by spin-rotation coupling. *Phys. Rev. B* **87**, 180402 (R) (2013).

- [20] Küß, M. et al. Nonreciprocal Dzyaloshinskii-Moriya magnetoacoustic waves. *Phys. Rev. Lett.* **125**, 217203 (2020).
- [21] An, K. et al. Coherent long-range transfer of angular momentum between magnon Kittel modes by phonons. *Phys. Rev. B* **101**, 060407(R) (2020).
- [22] Man, H. et al. Direct observation of magnon-phonon coupling in yttrium iron garnet. *Phys. Rev. B* **96**, 100406(R) (2017).
- [23] Casals, B. et al. Generation and imaging of magnetoacoustic waves over millimeter distances. *Phys. Rev. Lett.* **124**, 137202 (2020).
- [24] Li, Y. et al. Hybrid magnonics: Physics, circuits, and applications for coherent information processing. *J. Appl. Phys.* **128**, 130902 (2020).

Reviewer #2:

In their work entitled Long magnon decay length in BiFeO₃/La_{0.67}Sr_{0.33}MnO₃ heterostructures by J. Zhang et al., the authors report on the experimental observation of unprecedented long spin wave decay lengths up to 1 mm in a heterostructure comprised of a multiferroic BiFeO₃ and a La_{0.67}Sr_{0.33}MnO₃ layer underneath which is typically in the half-metallic ferromagnetic phase for this composition. The authors attribute this ultra-long decay length to the formation of a hybrid magnon-phonon mode (denoted X mode from the authors) which propagates with a much higher group velocity than the “normal” spin wave mode and an associated ultra-low relaxation rate of this hybridized state.

The authors used a set of different experimental methods such as angular-dependent (the angle between the wave-vector and the in-plane externally applied static magnetic field) propagative spin wave spectroscopy using different devices where they varied the distance between the two antennae to show the spin wave transmission about long distances. The authors used them as well as for the determination of the group velocity. These findings were confirmed by performing time-resolved Brillouin Light Spectroscopy as well. Following a theoretical model from Yamamoto et al. , the authors explain their findings by a simplified model where- via an isotropic magnetoelastic coupling- the spin wave modes in the LSMO film couple with a longitudinal acoustic wave mode. The experimental findings are in a good agreement with the simulations performed based on this model, correctly reproducing the most important findings.

Response:

We appreciate Reviewer #2 for his/her positive comment and very nice summary of our manuscript.

Comments on validity, clarity etc.

The experimental data and the associated theoretical model are well presented, and the utilized methods are all described as well. In my view, the overall data analysis and interpretation is sound and well justified. The article’s findings is original and the topic is timely and contributes to the interdisciplinary activities in the utilization of different hybridized systems, enriching various research areas beyond pure magnonics. The article is written in a clear and concise manner. The supplementary material is detailed and further elucidates the main statements of the paper.

However, although there is a good agreement with the experiment and the theory based on magnon-phonon hybridization, partially the proof of the existence of such a hybrid state might be not fully clear for the non-expert reader in the main text. Thus, I have some questions/remarks to the authors.

Response:

We thank Reviewer #2 for finding our work “**original and the topic is timely and contributes to the interdisciplinary activities in the utilization of different hybridized systems, enriching various research areas beyond pure magnonics**”. We response point-by-point to his/her questions/remarks in the following.

Questions and remarks to the authors

Naively, when speaking of a hybridized state between two distinct systems with different dispersions, I expect to see some sort of avoided level crossing spectra, even if the system is not fully strongly coupled, which would still be reflected in the corresponding reflection/transmission spectra. The authors devote the supplementary material section VI. to show calculated, not experimental, dispersion relations only in the supplementary material and in one figure in section IX. (Fig S11) of the same supplementary material. However, I cannot see that type of feature in Fig. S11 shown for specific k. Possibly a zoom to the anticrossing region might be helpful and answer already my concern. Additionally, as showing the existence of such hybrid mode is crucial for the statements of this work, I personally think it might be elucidating if this figure appears in the main text.

Response:

The zoom-in image near the coupling region is displayed below after further tuning the color-coding contrast. Here, we can observe some sort of anti-crossing features between the magnon mode (field dependent) and the phonon mode (field independent). Based on such experimental observation, we would not claim to fully enter the strong coupling regime. In the present systems, the coupling (estimated by comparing the experiment and simulation) seems to exceed *intrinsic* damping of magnons and phonons, which is the definition of strong coupling, but the *extrinsic* broadening arising from the measurement scheme is much larger than the coupling. This makes it impossible to see a clear anti-crossing feature. Nevertheless, these experimental data do reveal some sort of coupling or hybridization between magnons and phonons.

We appreciate this helpful suggestion of the reviewer. We fully agree and therefore have included the above figure as the new Fig. 3a. In the main text, we have added some discussions on this anticrossing-like feature observed in experiments as:

“When the k_3 phonon mode (field-independent branch) meets the magnon mode (field-dependent branch), the anticrossing-like feature is observed as shown in Fig. 3a revealing the magnon-phonon hybridization (white arrow). By rotating the in-plane magnetic field of 100 mT, the magnon-phonon hybridized mode forms mode X right above mode Y, as shown in Supplementary Fig. 13. Essentially, the long decay length of mode X arises from the magnon-phonon hybridization.”

Regarding that, the LA velocity of the mode in BFO is reported to be 2.5 km/s which is the same value as the one found for the X-mode. Hence, is there really a propagation of the once, by the first antenna, excited hybrid mode or do we have here the coupling between a propagating phonon which- by the magnetoelastic coupling- is “exciting magnons along the way”, such that it is seen by the utilized experimental methods probing the magnons? Could the authors please comment on that?

Response:

We thank Reviewer #2 for this interesting question. In our understanding, within the frequency range of order anti-crossing gap around the crossing point, one cannot distinguish between magnons and phonons regardless of the damping. Therefore, we call this hybridization. At the excitation region near the antenna, we consider that the hybridized mode has already formed. Since the CPWs (or NSLs) are designed to excite magnons, they are current-driven, not voltage-driven, and as a consequence, they excite magnons much more efficiently than phonons. At the frequency/field condition for mode X, the phonon mode is no longer pure phonon, but hybridized with magnons. However, as we discussed in the response to Reviewer #1, the longest propagating modes we observed might well be somewhat off the anti-crossing region. These phonons couple less strongly with magnons, and in this case, the “exciting magnons along the way” picture may be appropriate.

Note that if magnon relaxation frequency is still smaller than the resonance frequency shift of phonons due to the coupling, one may still talk about hybridization. If magnon relaxation is stronger, magnons excited by phonons are unable to come back to interact again with the phonons, preventing them from hybridization. To summarize, our answer to the question is that what we observe is a mixture of the hybrid modes and phonons leaking magnons away. We appreciate that it is an interesting conceptual question, but cannot offer quantitative analysis on it due to the limited experimental resolution and difficulty in systematic fitting of data by the theory.

Further, in the main text the authors write when discussing the hybridized state in line 177-178 “enhancement in the transmission spectra is observed” but in the supplementary material they also write in section VI. “[...] reflected upon the strong enhancement of the S_{22} signal which is not related to the magnon-phonon hybridization.” Could the authors please clarify?

Response:

We regret that this statement was confusing and blurring our main point. In the main text, we focus on the S_{21} (transmission) signal for BFO/LSMO samples, and we interpret the enhanced signal and decay length as arising from the magnon-phonon hybridization. In the supplementary, we also show the S_{22} (reflection) signal (now Fig. S13) in which we see a little similar signal enhancement near 45 degree. Since for reflection the propagation distance should not matter, the almost-flat dispersion at least partially explains the behaviours in Fig. S13b. We conclude that the magnon-phonon hybridization and the flat magnon dispersion both contribute to a strong mode X, particularly in the reflection spectra. We have now rephrased this discussion in the supplementary information as follows:

“This phenomenon may contribute to the strong enhancement of the S_{22} reflection signal (Fig. S13b, top row in Fig. S7) in addition to that of the magnon-phonon hybridization. However, the strong transmission S_{21} of mode X mainly results from the magnon-phonon hybridization since a flat magnon mode has zero group velocity. Instead, a magnon-phonon hybridized mode exhibits high group velocity and long life-time.”

It is known from cavity magnonics that the linewidths are changing towards the hybridization. Today, one distinguishes between coherent coupling with anticrossings as their hallmark and dissipative couplings with level crossings. It was shown that in the presence of dissipative coupling the linewidth of the hybrid state is decreased, and the amplitude increases. Furthermore, similar to some works in cavity magnonics (One tone approach from Harder et. al, PRL 2018 two tone approach, theory: Grigoryan et al. PRB 2018, experiment: Boventer et al., PRR 2020), the coupling strength becomes complex or purely imaginary as is the expression of the coupling strength from the authors in the supplementary material. I wonder if the authors have considered the possibility of such type of hybridization while seeing ‘only’ an enhancement in the spectra at the position of the conditions for the hybridization (Fig. S11) and if it could at least partially account for the observation of such ultra-low relaxation rate the authors reported. Could they please also comment on that?

Response:

This is a very insightful remark. Yes, we believe that the type of coupling at the crossing point may as well be level attraction. Nevertheless, from experimental results such as the newly added Fig. 3a, we cannot tell for sure whether it is anti-crossing or level attraction, simply because our measurement resolution is insufficient. We agree that the level attraction may result in line narrowing and enhancement of the signals as studied by previous works. Therefore, although we stay on the conservative side and put coupling type as anti-crossing in the simulation shown in the new Fig. 3b, we do not rule out the possibility of level-attraction type of coupling at the crossing point as shown in simulation results in the right figure below,

Note that the value of effective coupling $b_{\text{eff}} = 2 \times 10^7$ used here is way too large and would give an S_{21} simulation result inconsistent with the experimental data. Therefore, we have added a sentence in the later part of the main text to include this discussion as

“Although here we assume the coupling type to be anti-crossing, it may also be of level-attraction type [49] which may induce signal enhancement[50] and line narrowing[51] around the crossing point.”

[49] Harder, M. et al. Level attraction due to dissipative magnon-photon coupling. *Phys. Rev. Lett.* **121**, 137203 (2018).

[50] Grigoryan, V. L., Shen, K. & Xia, K. Synchronized spin-photon coupling in a microwave cavity. *Phys. Rev. B* **98**, 024406 (2018).

[51] Boventer, I. et al. Control of the coupling strength and linewidth of a cavity magnon-polariton. *Phys. Rev. Res.* **2**, 013154 (2020).

We also appreciate that the input-output theory perspective of Ref. [51] could offer a further insight into our experimental data. For instance, we could regard the CPW microwave mode as a dynamical entity, which couple with phonons via magnons. However, this analysis would require rewriting the theoretical part from scratch. Given that the present model already satisfactorily explains the data, we would like to leave it for a future study. Nonetheless, we revised Sec. VI in Supplementary to discuss anti-crossing and dissipative coupling.

What would be the lowest coupling strength possible to still see such long decay length?

Response:

If the dissipation is ignored, the anti-crossing theoretically occurs for any nonzero coupling. Thus, if the damping is zero and one has an infinite experimental resolution, there should always be a small frequency and field range in which the signal enhancement occurs. In reality, of course, the damping is nonzero, in which case the lowest value of coupling required is such that the resulting anti-crossing gap is greater than the linewidth. In fact, even if the coupling is not fully in a strong coupling regime, we still expect some sort of hybridization between magnons and phonons. Once the hybridization is formed, the phonon high velocity and long life-time will positively affect the magnon decay length. However, if the coupling is too weak, we doubt if we can still clearly observe the hybridized mode (mode X) in experiments. For example, when the magneto-elastic coupling is reduced to $b_{\text{eff}} = 2 \times 10^4$ in the simulation, as shown in the following figure, mode X vanishes with a propagation distance of 6 μm .

To be more precise, let us also point out that the simulation involves a fitting parameter c , which is taken $c = 6000$ for the main results and the figure here. This choice seems optimal for $b_{\text{eff}} = 2 \times 10^5$, but for other values of the coupling, one is free to adjust c to make the result look more like the

experiment. For weak couplings, one could indefinitely increase c to make the mode X visible. The honest situation is that we cannot really determine b_{eff} and c independently, but only their combination. In reality, c should be a material parameter, and a large c means a very sensitive piezoelectric effect. So there should be a minimal coupling strength required for observing the long decay length, but we are unable to pin it down quantitatively.

More discussion and relevant simulation results are shown in the revised Supplement Sec. VI.

In the following I only have minor comments.

a) What is the initial direction of the, for instance, out-of plane component, of the ferroelectric polarization state of the BFO layer? How does it affect the hybrid X mode if that state would be flipped, as it has been shown that the ferroelectric domain polarization state also can have an impact?

Response:

The initial direction of the out-of-plane component of the ferroelectric polarization is upward and we characterized with piezoelectric force microscopy (PFM), as shown in the following figure.

In our device with integrated CPWs or NSLs, the antennas are very fragile against the voltage applied by the PFM tips. Therefore, to study the spin-wave propagation while switching the electric polarization of BFO is not straight-forward in our experimental setup. However, we believe that further investigation of voltage-controlled spin-wave propagation in particular of the mode X would be very interesting, although it is beyond our current technique and out of the scope of the main focus of this work. Thanks to this helpful question from the reviewer, we became aware of a very recent work [Merbouche et al. *ACS Nano*, **15**, 9775-9781, (2021)] on voltage-controlled magnonic crystal demonstrated on relevant material system, and therefore we added one sentence in the outlook section to very briefly discuss about this point as

“In view of the multiferroic nature of BFO, future investigation may lead to voltage manipulation [52] of the magnon-phonon hybridized mode by switching the ferroelectric polarization.”

[52] Merbouche, H. et al. Voltage-controlled reconfigurable magnonic crystal at the sub-micrometer scale. *ACS Nano* **15**, 9775-9781 (2021).

b) The scale bars are only mentioned in the captions. I would suggest adding the scales rather to the image, helping to grasp the entire figure even faster.

Response:

We thank Reviewer #2 for this suggestion and have added the values of scale bars in all figures in the updated version of the main text as well as the supplementary information.

c) What is the orientation of the NGO substrate, 001 or 110, for instance?

Response:

The out-of-plane crystalline orientation of the NGO substrate used in this work is [001].

d) Line 258: The value of 29 GHz/T for the gyromagnetic ratio is a fit result and hence not 28 GHz/T?

Response:

Yes, the PPMS with phaseFMR module automatically yields a value of 28.5 GHz/T for the gyromagnetic ratio from fitting of the FMR measurement presented in Fig. S3. Therefore, we have used roughly 29 GHz/T in the simulation, as we can of course also use 28 GHz/T, which will not make much influence on the simulation results, we believe.

Recommendation:

In conclusion the observation of such a long decay length marks a significant advance for magnonics and beyond. The utilization of such hybridized mode opens new routes for the integration of magnonic devices to other architectures and opens new research directions based on such BFO/LSMO heterostructures as it can merge the functionalities of hybridized states and functional oxides together.

Thus, in principle, I can recommend this work for publication after minor revision in Nature Communications if the nature and its experimental demonstration of the hybridized state is further clarified.

Response:

We thank Reviewer #2 for his/her positive comment **“the observation of such a long decay length marks a significant advance for magnonics and beyond”** and the recommendation for publication.

Reviewer #3:

The paper presents experimental evidences of waves propagating in BFO/LSMO bilayer deposited on a NGO substrate. Two modes are evidenced. The slower one is the dipole exchange mode propagating perpendicularly to the magnetization direction in LSMO. The faster one propagates with a group velocity of 2.5 km/s. This latter one is interpreted as an acoustic wave propagating in BFO. The experimental results are nice but the interpretation is questionable (see the attached file). I suggest major revisions before publication.

Response:

We thank Reviewer #3 for having nicely summarized the manuscript.

The experimental results are nice but the interpretation is questionable because 2.5 km/s corresponds to shear wave velocity in LSMO [Phys. stat. sol. (a) 195, No. 2, 350–358 (2003)]. According to [Appl. Phys. Lett. 102, 182905 (2013)] the BFO shear elastic constant C_{66} is 89 GPa. Taking a BFO mass density of 8220 kg/m³, we obtain a shear wave velocity of 3.3 km/s. According to [Eur. Phys. J. B 94, 108 (2021)] the NGO shear modulus is 90 GPa. Taking a NGO mass density of 7.3 kg/m³, we obtain a shear wave velocity of 3.5 km/s. According to these estimations, the LSMO shear wave velocity is lower than those in BFO and NGO. This means that LSMO sandwiched between BFO and NGO is an acoustic wave guide. Consequently the observed wave propagating at 2.5 km/s is localized within LSMO. Coupling acoustic waves and dipole exchange waves within LSMO is more likely than coupling waves in different media BFO and LSMO.

Response:

We thank the reviewer for raising this question. First, if the coupling were between phonons and magnons within LSMO only, we should be able to observe long-distance propagation of mode X in pure LSMO sample. However, from Fig. S4b, we can see that mode X is almost invisible in the transmission spectra. This would suggest that the coupling with BFO is necessary to form the hybridized mode X. Second, we note that the pure phonons in BFO may have slightly higher group velocity than the observed mode in the experiments. As mentioned in the response to Reviewer #1, we measured the sound velocity for a pure BFO sample and obtained 3.2 km/s. This may result from its hybridization with magnons whose group velocity is much slower, below 1 km/s. This is in fact an indirect evidence that the observed long-distance mode has the nature of not pure phonons, but rather the magnon-phonon hybridized mode. Third, we also remark that hybridization between magnons and TA phonons should have a different dependence on the magnetic field direction than the hybridization with LA phonons.

We thank the reviewer for pointing out reference [Appl. Phys. Lett. 102, 182905 (2013)]. Based on their numerical model calculation, the phonon velocity in BFO can be estimated to be 3.3 km/s which is higher than the TA phonon velocity 2.2 km/s reported experimentally by neutron scattering in Ref. [33]. Furthermore, 550 m/s was obtained for surface mode by a transmission experiment in Ref. [25]. If the surface mode is a Rayleigh wave, which is quite probable here, its velocity is only up to around 10% smaller than the TA velocity, which suggests a value significantly lower than 3 km/s. Generally, therefore, the numerical results of [Appl. Phys. Lett. 102, 182905 (2013)] seem to overestimate the elastic constants in comparison with the experimental data of [25,33]. Still, all these results point to LA sound velocity of order 3 km/s for BFO, which seems at least consistent with our interpretation considering the uncertainties in all the experiments as well as in the numerics. We have now included these discussions in the main text as

“The magnon-phonon coupling within LSMO cannot account for the hybridized mode X in view of the fact that mode X vanishes in the transmission spectra on a bare LSMO film without BFO (see Supplementary Fig. 4). A neutron scattering experiment [33] reported the velocity of longitudinal phonon mode to be 2.6 km s^{-1} , which is fairly close to the value (3.2 km s^{-1}) estimated from our experiments on a pure BFO sample (see Supplementary Fig. 14). Therefore, we assign mode X to the BFO phonon mode hybridized with LSMO magnon mode. The velocity of mode X (2.5 km s^{-1}) is slightly lower than that of phonons in pure BFO [33,46] due to the hybridization with the slow magnon mode.”

[46] Dong, H., Chen, C., Wang, S., Duan, W. & Li, J. Elastic properties of tetragonal BiFeO₃ from first-principles calculations. *Appl. Phys. Lett.* **102**, 182905 (2013).

Concerning the model, could the authors explain why the exchange bias field has the same in-plane and out-of-plane contributions and why the in-plane contribution depends not on magnetization direction?

Response:

We regret that this caused some confusions. The constant B was introduced as a convenient fitting parameter since the precise modelling of all the magnetic free energy contributions in such a complicated heterostructure is impossible. We are not sure about the origin of B, but one possibility could be that the antiferromagnetic Néel vector rotates as the in-plane magnetic field is rotated (possibly by minimising the magneto-static energy arising from the weak ferromagnetism of BFO), in which case the exchange bias should appear isotropic for the LSMO magnetization. However, this is just one possibility and we do not have any supporting evidence, so that we have removed this remark and simply introduced B as a fitting parameter in the revised manuscript.

REVIEWER COMMENTS

Reviewer #1 (Remarks to the Author):

In the revised manuscript, the authors properly answered most of my questions/comments, some of them are still not very clear yet. I can recommend the manuscript for publication in Nature Communications after clearly addressing the rest problems.

(1) The authors experimentally demonstrate a long propagation length of the acoustic wave in BFO using IDT (Fig. S14). However, the Q-factor of the mode at $f \approx 1.8$ and 5 GHz is low (<50). Since the frequency and linewidth of an acoustic wave is related to the IDT geometry, can the authors provide more information on the IDT such as electrode width, period, and number of electrodes? Does the mode of $f \approx 5$ GHz correspond to a harmonic mode? Can the authors also comment on the low-Q?

(2) In BFO/LSMO, long decay length of the mode X is attributed to the coupling of the magnon to the phonon, i.e., the low-loss acoustic wave carrying the magnon propagates over 1 mm. Going back to my question about the coupling for $f > 6$ GHz (Question 4), the authors argued the coupling of magnons at high frequency to the high-order phonon modes e.g., the k4 mode, however, the signal of the mode X seems to be stronger with the magnetic field increasing (see Fig. S18b and d). As the k4 phonon mode is weakly excited by the CPW, one expects to observe the weaker mode X at the higher frequency. By comparing the data in Fig. S13d and Fig. S18, the phonon intensity measured by the CPW seems to be different, i.e., the phonon modes are not observed in Fig. S18. Can the authors comment on those? In addition, can the authors evaluate theoretically the coupling strength extracted from the anti-crossing gap for different phonon modes, e.g., k1 - k6 (see main Fig. 3b). In the experiment, thermal phonons of high frequency may play a role in the coupling. This may explain why the coupling occurs in a wide frequency range.

(3) In BLS measurements, the authors mention that the decay of the mode X in the 2D spatial map, particularly in the first few micrometers, is primarily determined by the magnon decay length. Can the authors present a 2D image with a large y range where the hybridized mode X can be visualized?

(4) The title of the manuscript is "long magnon decay length in BiFeO₃/La_{0.67}Sr_{0.33}MnO₃ heterostructures". Since long decay length doesn't apply for pure magnons in BFO/LSMO, to not be confused, the author may think about a different title like "Long decay length of magnon-polarons in BiFeO₃/La_{0.67}Sr_{0.33}MnO₃ heterostructures".

Reviewer #2 (Remarks to the Author):

I would like to thank the authors for their detailed response on the other reviewers and my own remarks to their manuscript on the long magnon decay length in BFO/LSMO heterostructures. The authors have mostly answered my concerns and together with the other reviewers comments I also believe that the manuscript has much improved. However, there is one main point remaining which still needs to be clarified in my view (Please also have a look on my comments in the attached file).

The experimental data is well presented and definitely interesting. The authors have greatly improved the presentation and discussion of these in the main part of the manuscript.

However, to me the possible underlying physical mechanisms for that observation of such a long decay length- are listed but its not clear what is the dominating one governing the propagation. Correspondingly (also see my comments) I don't consider the answer to my "excitation of magnons in the fly" sufficient to explain the main source of the long decay length (also in combination with the 2D map provided now by the authors)/ and the one to anticrossings/level attraction. Could the authors

please further comment on that?

Based on this, I still could recommend the manuscript for publication in Nature Communications, if this answer is answered more clearly as it is indeed an important question in (hybrid) magnonics.

Yours sincerely,
Reviewer

Reviewer #3 (Remarks to the Author):

the novelty of the experimental data and the effort carried out for addressing the issues raised in the original version make the present version suitable for publication in Nature Communication

Response to referee reports

First of all, we thank all three reviewers for their time and efforts and for their in-depth comments and suggestions which we find very constructive and helpful. We took them into account fully and carefully revised the manuscript which we believe has significantly improved. We are therefore optimistic that the manuscript now meets the requirements for publication in *Nature Communications*. Below we provide our point-to-point responses to all of the reviewer comments.

Reviewer #1:

In the manuscript, Zhang et al., present the experimental results on a magnon decay length of up to one millimeter in multiferroic/ferromagnetic BiFeO₃(BFO)/La_{0.67}Sr_{0.33}MnO₃(LSMO) heterostructures using angle-resolved propagating spin-wave spectroscopy and time-resolved Brillouin light scattering. The authors reveal the nature of ultra-long magnon decay length based on the magnon-phonon coupling at the interface of BFO/LSMO. The high group velocity extracted from the experiments and long relaxation time assumed in the manuscript are required to attain 1-mm decay length. I find these experimental results are interesting and could have an impact on hybrid magnonics, which is rapidly growing. However, in my opinion, it is not convincing that the long decay length of magnons in BFO/LSMO arises from the magnon-phonon coupling and therefore, more experimental and/or theoretical data are needed to make a solid conclusion.

Response:

We thank Reviewer #1 for finding our work “**interesting and could have an impact on hybrid magnonics**”. As mentioned, we have done additional experimental and theoretical work to further consolidate this conclusion, as detailed in the following.

(1) The decay length of an acoustic wave at GHz frequency is typically in the millimeter range, as written in the manuscript. In a hybrid system, e.g., Refs. 14 and 29, magnons can benefit from the hybridization with long-lifetime phonons, resulting in long-distance propagation. In the manuscript, I didn't find the authors show the experimental evidence for long-lifetime acoustic phonons in BFO. Instead, they mentioned some references (e.g., Refs. 24 and 25) about the phonon lifetime. From these references, I cannot conclude the acoustic phonons of 2-6 GHz in BFO host very long lifetime that allows for a propagation distance of 1 mm. The authors should add more relevant data to address this point.

Response:

In order to address this point, we have prepared a new sample of pure BFO and integrated two interdigital transducers (IDTs) to excite and detect phonons over different distances being 50 μm and 1 mm, respectively.

One can see from the transmission spectra S_{21} that the signal strength for the $d=1$ mm sample is only slightly weaker than that of the $d=50$ μm sample, which suggests that the decay length of phonons in BFO is approximately 1.7 mm with a rough estimate. The phonon velocity in pure BFO is also estimated to be 3.2 km s^{-1} , which is slightly higher than 2.6 km/s quoted in Ref. [32] and 2.5 km/s in our main data. The difference may be attributed to the effect of substrate, different crystalline orientations, and the hybridization with magnons, as discussed below in response to Reviewer #3.

We have now included these additional experimental data in the supplementary information as Section X, and added a sentence in the main text “...as experimentally demonstrated in a bare BFO sample with integrated IDTs (see Supplementary Fig. 14).”

(2) Let’s assume that the acoustic phonons with a long lifetime in BFO strongly couple to the magnons in LSMO. According to the formula of $\lambda = 2vg/\Delta\omega$ written in the manuscript, the minimum value of $\Delta\omega$ for a hybridized mode can be simply estimated by $(\Delta\omega_{\text{magnon}} + \Delta\omega_{\text{phonon}})/2$, which is dominated by $\Delta\omega_{\text{magnon}}$. With considering enhanced group velocity vg , the magnon decay length in BFO/LSMO will increase by one order of magnitude, rather than $>$ two orders of magnitude stated in the manuscript. Can the authors comment on this argument?

Response:

We thank Reviewer #1 for this very insightful discussion. In a very ideal (theory/simulation) scenario where all parameters (i.e. the frequency f , the wavevector k and also the magnetic field H) are finely tuned to reach a crossing point, the anti-crossing (or level-repulsion) should occur, which could be observed with a fixed H and varying k , and alternatively with a fixed k and varying H as shown in the simplified simulation results considering the coupling of magnon and phonon branches. Indeed, in either case, at the crossing point, the linewidth should be only slightly narrower than that of pure magnons.

One trivial issue for simulation but non-trivial issue for experiments, is how to efficiently excite magnon and/or phonon with high wavevector k of about 20 $\text{rad}/\mu\text{m}$. In all our experiments, we have to admit that we did not manage to efficiently excite modes with wavevectors of up to 20 $\text{rad}/\mu\text{m}$ as one may see from the wavevector profile of the CPW calculated from Fourier transformation as shown in the red dashed line in the newly added Fig. 3b (we copy it here for the convenience) and likewise for nanostriplines (Figs. S2g and h). However, the high-order wavevector k_3 (green circle) is located near the crossing point and the excitation can no longer be considered as 100% pure phonons, but instead contains a magnetic (magnon) information. In other words, the mode X observed experimentally may well be the “green circle” mode being a hybridized mode off the crossing point, which contains magnon information and yet, almost fully retains its phonon high velocity and long life-time.

The key for the observation of the “green circle” mode is the great sensitivity of the piezoelectric effect, i.e. $D = d * T$ where D is the electric flux density, T is the stress, and d is the piezoelectric coefficient, which is very large indeed. Thus, a small stress generates a large voltage. Instead, even a large current can cause only a minor deformation to the crystal. This has two consequences:

1. On the excitation antenna, the rf current mainly drives magnetization dynamics while the direct excitation of acoustic waves is very small. The rf current drives non-resonant magnons and the magnons generates an effective force field on the phonons owing to the magneto-elastic coupling. The phonon amplitude can be estimated as $A_{\text{phonon}} \approx \frac{g_{\text{eff}}}{\Delta\omega} A_{\text{magnon}}$, where g_{eff} is the coupling strength and $\Delta\omega$ is the difference in the resonance frequencies. Even when the phonon and magnon are separated by 1GHz, there is still a 5% hybridization between them. The main part of the phonon signal in our experiments comes from this magnon-phonon hybridization, not from the inverse piezoelectric excitation by the rf current.
2. On the receiving antenna, in particular at large distances, the signal is dominated by the direct piezoelectric effect. This is consistent with the fact that the long-distance signals do not show phase oscillation (the signal is always entirely negative). This antenna picks out the phonon-dominated signal which is weakly hybridized with magnons. Still the signal is distinctively magnetic in that it depends strongly on the magnetic field direction.

In summary, the mode X observed in experiments, including the magnetic field dependence and long decay length, can be attributed to a mildly hybridized mode (slightly away from the crossing point) corresponding to the magnon-phonon coupling, as indicated by the green circle in the newly added Fig. 3b showing the dispersion relations.

(3) In Fig. S11d, the authors attribute three absorption dips to the acoustic phonons. From the data, I find the phonon linewidth, related to $\Delta\omega_{\text{phonon}}$, is quite large. It seems that the phonon lifetime is rather short, which conflicts with the statement in the manuscript. Can the authors comment on this? In addition, please describe the excitation mechanism of phonons using CPWs.

Response:

We thank Reviewer#1 for his/her careful observation on the linewidth of phonon mode in supplementary Fig. S11d (now Fig. S13d).

In these data, the broad linewidth of the phonon signal does not arise from the intrinsic damping of phonons, but is rather dominated by the $\Delta\omega$ induced by broad wavevector excitation Δk of the CPW via $\Delta\omega = v_g * \Delta k$. Since phonons have relatively large group velocity v_g , a finite Δk (determined by the width and imperfections of the CPW antennas) will produce a large $\Delta\omega$, much larger than the linewidth of intrinsic phonon lifetime.

To have a rough estimate of this inhomogeneous broadening, we read Δk from Fig. S2i to be roughly about 2 rad/ μm . Taking the phonon velocity of roughly 3 km/s, this would yield a linewidth broadening $\Delta\omega=v_g*\Delta k$ to be about 1 GHz. To clarify this point in the manuscript, we have added one sentence in the main text: “*The phonon modes exhibit broad frequency band owing to the broad wavevector excitation and high phonon velocity.*”

We agree with the reviewer that the CPWs are designed for the purpose of exciting and detecting magnon signals. Therefore, in most cases, we do not observe clear phonon signals. Nevertheless, since CPWs are directly deposited on top of the multiferroic BFO, in some cases the CPWs may generate oscillating electric field and act as incompetent (or shorted) interdigit transducers (IDTs) that may trigger some phonon modes in BFO via piezoelectric effects.

(4) The 1-mm magnon decay length was extracted from Fig. S15. The signal is weak for $d > 100 \mu\text{m}$. I see a bit stronger signal at a magnetic field $> 100 \text{ mT}$, especially for $d = 940$ and $1060 \mu\text{m}$. The spin-wave frequency is larger than 6 GHz for the field strength $> 100 \text{ mT}$. In Fig. S11d, the phonon frequency is below 6 GHz and is magnetic field independent. How does the magnon-phonon coupling occur for a magnetic field $> 100 \text{ mT}$?

Response:

We thank the reviewer for pointing out this observation. Below we put the data in supplementary Fig. S11d (now Fig. S13d) with $d=12 \mu\text{m}$ as (a) together with those for $d=940 \mu\text{m}$ as (d) for comparison. We can see that the enhancement around 7 GHz, clearly visible in (a) still exists in (d). So this main feature of the data is consistent at all distances. We agree with the reviewer that there does exist some sizable signal above 100 mT and around 10 GHz. This additionally observed mode may be due to even higher order excitation of the CPW antenna. In fact, we checked the two meander CPWs patterned on these two samples (b and e) by SEM and found the widths of the antenna signal and ground lines to be slightly different, due to the lithography process. Such mild differences may have little impact on the k_1 mode, but may impact significantly the high order excitations, such as k_3 and k_4 as shown in the Fourier transformation results, as shown in (c) and (f). The signal above 100 mT around 10 GHz may be associated with the k_4 excitation of the CPWs due to a slight change in the width of the conducting lines.

Furthermore, the Fourier transform of the antenna geometry is just a crude approximation to the real wavenumber excitation spectrum. Phonons and magnons are driven by the electric and magnetic fields respectively generated by the rf current in the CPW. And the field profiles should show some extra spatial dependence in the electric field arising from the charge distribution inside the antenna and in the magnetic field owing to the $1/r$ dependence of Oersted field. As explained in our response to point (2) of this referee, the main driving force is magnetic in this experiment. Indeed, the old Fig. S11d showed that the pure phonon excitation is cut-off above 7GHz whereas the magnetic signal is clearly visible up to 12 GHz in old Fig. S11d too. Consistent with our discussion of point (2) above, at these frequencies far away from the crossing point, the magnetization does not excite significant phonons via magneto-elastic effect.

We find these observations in comparing data sets useful and therefore added one sentence in the supplementary information to comment on this as:

“We note that the real CPW excitation spectrum is further broadened from Fig. S2i due to imperfections in fabrication, and spatially inhomogeneous profiles of the induced electromagnetic fields. Thus, the transmission signal above 100 mT and around 10 GHz may be associated with higher order excitation of the antennas.”

(5) The authors analyzed magnon-phonon coupling based on a numerical model. They assume the coupling only exists at the interface. What is the thickness of the interface ($t_{\text{interface}}$)? The interfacial coupling may lead to a difference in the S12 and S21 spectra. The effect might be small since the measurement geometry is away from the Damon-Eshbach configuration. Can the authors compare S21 to S12 measured in BFO/LSMO?

Response:

In the theoretical model in this work, $t_{\text{interface}}$ appears only through the combination shown in Eq. (S4). Since this is also the only place where the “bare” magneto-elastic coupling constant b appears, and since both $t_{\text{interface}}$ and b are fitting parameters of the model, one can estimate only b_{eff} from the comparison with the experimental data, but not $t_{\text{interface}}$ and b individually. The numerical results on Figs. 2f-h and Fig. S7 were produced for $b_{\text{eff}} = 2 \cdot 10^5$. It is impossible to even guess the real value of b , but if we take $b = 2 \cdot 10^6$, we would get $t_{\text{interface}} = 4$ nm. The smaller b is, the larger is $t_{\text{interface}}$.

The question of asymmetry between S12 and S21 is interesting and important, but out of scope of the present study that focuses on the magnon propagation distance. The theoretical model is too simplified for the asymmetric magneto-elastic coupling between the top and bottom surfaces to be incorporated. Specifically, we approximate the spin wave dynamics by averaging its spatial profile over the thickness direction so that there is no distinction between the Damon-Eshbach modes localized on the top and the bottom surfaces.

In the experimental data, the mode Y is dominated by Damon-Eshbach contributions and they clearly show some asymmetry between S12 and S21, as shown in the figure below.

Therefore, the difference exists, but it does not seem to affect the mode X significantly, maybe because they contain less of the Damon-Eshbach contribution as suggested by the reviewer. We would like to leave it for some future studies. 
(6) The intensity of the mode X calculated by the numerical model (Fig. 2f-h) doesn't rely on the field angle. However, the mode X should vanish near the field angle of 0 and 90 degree since the coupling coefficient (equation S2) is nearly zero. Can the authors comment on this?

Response:

As the reviewer pointed out, the magneto-elastic coupling coefficient is zero for 0 and 90 degree, and the numerical calculation does reflect this correctly. In the original manuscript, we plotted Figs. 2f-h in the angular range between 30 and 150 degrees so that the vanishing of mode X at 0 is not shown, but it is shown to disappear at 90 degrees. In Fig. S7, the mode X is shown to vanish at 0 and 90 degrees. In the revised manuscript, we have changed the angular range of Figs. 2a-h to 0-180 in order to avoid the confusion. The updated Fig. 2 is shown below.

(7) In Fig. S8, the dispersion relations of the phonon and magnon mode are calculated for $\alpha = 0$. In this case, strong magnon-phonon coupling occurs for $\theta \approx 45$ degree and the mode profile exchanges after the crossing point. However, for $\theta = 0$ and 90 degree, the magnon-phonon coupling should be weak. I expect no exchange between the phonon and magnon mode. In addition, if the authors take the experimental value of α , the anti-crossing gap is still clear. If yes, I suggest the authors include this part in the main manuscript and discuss the magnon-phonon coupling strength.

Response:

We agree with the reviewer's comment and there is no anti-crossing at 0 and 90 degrees, as can be seen in the insets of Fig. S8. The change of color from blue to orange does not indicate an exchange of

magnon and phonon mode, especially at zero anti-crossing at 0 and 90 degrees. Even at the anti-crossing, it is hard to fully distinguish the magnon and phonon mode, since the anti-crossing gap is small compared to the experimental k resolution. To avoid this confusion, we have now made a new Fig. S8 in which the color does indicate the magnon-phonon ratio.

In the simulation, we set a realistic value for the hybridization parameter of $b_{\text{eff}} = 2 \times 10^5$, which does not ensure strong coupling considering the experimental value of alpha as suggested by the reviewer. If we further increase the b_{eff} , it can certainly enter the strong coupling regime as shown in Supplementary Sec. VI. We have now included additional simulation results in the new Fig. 3b as well as Supplementary Sec. VI and added a few sentences in the main text to discuss on this as below.

“Figure 3b shows the simulation results on the dispersion relations where magnon and phonon branches are present. In this simulation, we set the hybridization parameter $b_{\text{eff}} = 2 \times 10^5$ (see Supplementary Sec. VI), which does not fully enter the strong coupling regime [34].”

[34] Berk, C. et al. Strongly coupled magnon-phonon dynamics in a single nanomagnet. *Nat. Commun.* **10**, 2652 (2019).

(8) The authors mention that the phase-insensitive behavior of the mode X may be related to the coupling with phonons. I am not sure the authors clearly explain this in the text.

Response:

We thank the reviewer to point this issue out. Using CPWs to excite and detect spin waves based on PSWS technique can routinely give phase information of propagating spin waves (as those in mode Y). However, it seems to us that using similar technique with IDTs to study propagating acoustic waves does not necessarily provide the same clear phase oscillations in the transmission signals. In previous studies (Refs. [15] and [20]) on propagating acoustic waves in LiNbO₃, one does not observe clear phase oscillations as we do for propagating spin waves.

“This phase-insensitive behavior might be due to the coupling with phonons since the PSWS technique seems not to detect phonon phase information even with IDTs [15,20].”

[15] Xu, M. et al. Nonreciprocal surface acoustic wave propagation via magneto-rotation coupling. *Sci. Adv.* **6**, eabb1724 (2020).

[20] Küß, M. et al. Nonreciprocal Dzyaloshinskii-Moriya magnetoacoustic waves. Phys. Rev. Lett. 125, 217203 (2020).

(9) Did the authors measure a 2D spatial map of the mode X using micro-BLS? If yes, it is better to include the data in the manuscript.

Response:

We thank the reviewer for this great suggestion. We have now added the 2D spatial map of the mode X measured by micro-BLS as Supplementary Fig. 19. The 270 nm-wide NSL antenna (yellow bar) is located at $Y=0$. The external magnetic field is set at 100 mT with $\theta = 48^\circ$ and the frequency is set at 7.5 GHz so that the spatial distribution of mode X is selectively monitored. Due to the long data acquisition time of the BLS setup, the spatial map is taken within a $6 \mu\text{m} \times 20 \mu\text{m}$ square region. One could observe the decay of mode X in the 2D spatial map, particularly in the first few micrometers, where the decay is primarily determined by the magnon decay length.

In the main text, we have now inserted one sentence to refer to these BLS spatial mapping data as

“In the near-antenna region, mode X experiences a fast decay attributed to the pure magnon contribution as shown in Fig. 5 and the BLS spatial mapping near the NSL (see Supplementary Fig. 19).”

(10) The S21 spectrum of the mode Y shows much better oscillation behavior in BFO/LSMO (Fig. 1d) than that in pure LSMO (Fig. S4). Is there a physical reason behind?

Response:

We thank the reviewer for pointing out that the magnon signal in the LSMO sample without BFO is relatively weak particularly in transmission spectra. We attribute this to the difference in the quality of samples and integrated antennas rather than to physical reasons. This sample was produced at the start of this study. In view of this discrepancy, we have grown a new pure LSMO thin film with newly fabricated NSLs on top and replaced the previous results in Fig. S4 with the new measurement data as shown below.

The data on the new pure LSMO sample shows qualitatively the same behaviour except for a much stronger oscillation signal. The distance between two NSLs is $2\ \mu\text{m}$ and the applied field is fixed at 50 mT and rotated from 0 to 180 degree.

(11) The introduction does not cover sufficiently the field of magnon-phonon coupling and hybrid magnonics. Since the manuscript is going to demonstrate strong magnon-phonon coupling, I suggest the authors introduce more information on these topics.

Response:

We have followed the constructive suggestion of the reviewer to reorganize the introduction and inserted several sentences to better introduce the background information on magnon-phonon coupling and hybrid magnonics. The introduction now includes the following:

“Ferromagnetic resonance can be driven by surface acoustic waves (SAWs) via magnon-phonon coupling in a thin magnetic film on a piezoelectric substrate [13]. In the coupling process, phonon may carry spin information [14]. Nonreciprocal propagation of SAWs [15-20] is generated by the magnon-phonon coupling. Long-range transfer of angular momentum between two YIG layers separated by a thick substrate [21] is realized by the magnon-phonon coupling [22]. Direct imaging and quantification of both standing and propagating acoustomagnetic waves was presented in hybrid structures consisting of a piezoelectric substrate underneath Ni [23]. This way, magnon-phonon coupling and excitation by a large-area interdigital transducer (IDT) allowed to transport spin information over a macroscopic distance at specific IDT frequencies [23]. For future hybrid magnonics[24]...”

[13] Weiler, M. et al. Elastically driven ferromagnetic resonance in nickel thin films. *Phys. Rev. Lett.* **106**, 117601 (2011).

[14] Holanda, J., Maior, D. S., Azevedo, A. & Rezende, S. M. Detecting the phonon spin in magnon-phonon conversion experiments. *Nat. Phys.* **14**, 500-506 (2018).

[15] Xu, M. et al. Nonreciprocal surface acoustic wave propagation via magneto-rotation coupling. *Sci. Adv.* **6**, eabb1724 (2020).

[16] Sasaki, R., Nii, Y. & Onose, Y. Magnetization control by angular momentum transfer from surface acoustic wave to ferromagnetic spin moments. *Nat. Commun.* **12**, 2599 (2021).

[17] Yamamoto, K., Yu, W., Yu, T., Puebla, J., Xu, M., Maekawa, S. & Bauer, G. Non-reciprocal pumping of surface acoustic waves by spin wave resonance. *J. Phys. Soc. Jpn.* **89**, 113702 (2020).

[18] Maekawa, S. & Tachiki, M. Surface acoustic attenuation due to surface spin wave in ferro- and antiferromagnets. *AIP Conf. Proc.* **29**, 542 (1976).

[19] Matsuo, M., Ieda, J., Harii, K., Saitoh, E. & Maekawa, S. Mechanical generation of spin current by spin-rotation coupling. *Phys. Rev. B* **87**, 180402 (R) (2013).

-
- [20] Küß, M. et al. Nonreciprocal Dzyaloshinskii-Moriya magnetoacoustic waves. *Phys. Rev. Lett.* **125**, 217203 (2020).
- [21] An, K. et al. Coherent long-range transfer of angular momentum between magnon Kittel modes by phonons. *Phys. Rev. B* **101**, 060407(R) (2020).
- [22] Man, H. et al. Direct observation of magnon-phonon coupling in yttrium iron garnet. *Phys. Rev. B* **96**, 100406(R) (2017).
- [23] Casals, B. et al. Generation and imaging of magnetoacoustic waves over millimeter distances. *Phys. Rev. Lett.* **124**, 137202 (2020).
- [24] Li, Y. et al. Hybrid magnonics: Physics, circuits, and applications for coherent information processing. *J. Appl. Phys.* **128**, 130902 (2020).

Reviewer #2:

In their work entitled Long magnon decay length in BiFeO₃/La_{0.67}Sr_{0.33}MnO₃ heterostructures by J. Zhang et al., the authors report on the experimental observation of unprecedented long spin wave decay lengths up to 1 mm in a heterostructure comprised of a multiferroic BiFeO₃ and a La_{0.67}Sr_{0.33}MnO₃ layer underneath which is typically in the half-metallic ferromagnetic phase for this composition. The authors attribute this ultra-long decay length to the formation of a hybrid magnon-phonon mode (denoted X mode from the authors) which propagates with a much higher group velocity than the “normal” spin wave mode and an associated ultra-low relaxation rate of this hybridized state.

The authors used a set of different experimental methods such as angular-dependent (the angle between the wave-vector and the in-plane externally applied static magnetic field) propagative spin wave spectroscopy using different devices where they varied the distance between the two antennae to show the spin wave transmission about long distances. The authors used them as well as for the determination of the group velocity. These findings were confirmed by performing time-resolved Brillouin Light Spectroscopy as well. Following a theoretical model from Yamamoto et al. , the authors explain their findings by a simplified model where- via an isotropic magnetoelastic coupling- the spin wave modes in the LSMO film couple with a longitudinal acoustic wave mode. The experimental findings are in a good agreement with the simulations performed based on this model, correctly reproducing the most important findings.

Response:

We appreciate Reviewer #2 for his/her positive comment and very nice summary of our manuscript.

Comments on validity, clarity etc.

The experimental data and the associated theoretical model are well presented, and the utilized methods are all described as well. In my view, the overall data analysis and interpretation is sound and well justified. The article’s findings is original and the topic is timely and contributes to the interdisciplinary activities in the utilization of different hybridized systems, enriching various research areas beyond pure magnonics. The article is written in a clear and concise manner. The supplementary material is detailed and further elucidates the main statements of the paper.

However, although there is a good agreement with the experiment and the theory based on magnon-phonon hybridization, partially the proof of the existence of such a hybrid state might be not fully clear for the non-expert reader in the main text. Thus, I have some questions/remarks to the authors.

Response:

We thank Reviewer #2 for finding our work “**original and the topic is timely and contributes to the interdisciplinary activities in the utilization of different hybridized systems, enriching various research areas beyond pure magnonics**”. We response point-by-point to his/her questions/remarks in the following.

Questions and remarks to the authors

Naively, when speaking of a hybridized state between two distinct systems with different dispersions, I expect to see some sort of avoided level crossing spectra, even if the system is not fully strongly coupled, which would still be reflected in the corresponding reflection/transmission spectra. The authors devote the supplementary material section VI. to show calculated, not experimental, dispersion relations only in the supplementary material and in one figure in section IX. (Fig S11) of the same supplementary material. However, I cannot see that type of feature in Fig. S11 shown for specific k. Possibly a zoom to the anticrossing region might be helpful and answer already my concern. Additionally, as showing the existence of such hybrid mode is crucial for the statements of this work, I personally think it might be elucidating if this figure appears in the main text.

Response:

The zoom-in image near the coupling region is displayed below after further tuning the color-coding contrast. Here, we can observe some sort of anti-crossing features between the magnon mode (field dependent) and the phonon mode (field independent). Based on such experimental observation, we would not claim to fully enter the strong coupling regime. In the present systems, the coupling (estimated by comparing the experiment and simulation) seems to exceed *intrinsic* damping of magnons and phonons, which is the definition of strong coupling, but the *extrinsic* broadening arising from the measurement scheme is much larger than the coupling. This makes it impossible to see a clear anti-crossing feature. Nevertheless, these experimental data do reveal some sort of coupling or hybridization between magnons and phonons.

We appreciate this helpful suggestion of the reviewer. We fully agree and therefore have included the above figure as the new Fig. 3a. In the main text, we have added some discussions on this anticrossing-like feature observed in experiments as:

“When the k_3 phonon mode (field-independent branch) meets the magnon mode (field-dependent branch), the anticrossing-like feature is observed as shown in Fig. 3a revealing the magnon-phonon hybridization (white arrow). By rotating the in-plane magnetic field of 100 mT, the magnon-phonon hybridized mode forms mode X right above mode Y, as shown in Supplementary Fig. 13. Essentially, the long decay length of mode X arises from the magnon-phonon hybridization.”

Regarding that, the LA velocity of the mode in BFO is reported to be 2.5 km/s which is the same value as the one found for the X-mode. Hence, is there really a propagation of the once, by the first antenna, excited hybrid mode or do we have here the coupling between a propagating phonon which- by the magnetoelastic coupling- is “exciting magnons along the way”, such that it is seen by the utilized experimental methods probing the magnons? Could the authors please comment on that?

Response:

We thank Reviewer #2 for this interesting question. In our understanding, within the frequency range of order anti-crossing gap around the crossing point, one cannot distinguish between magnons and phonons regardless of the damping. Therefore, we call this hybridization. At the excitation region near the antenna, we consider that the hybridized mode has already formed. Since the CPWs (or NSLs) are designed to excite magnons, they are current-driven, not voltage-driven, and as a consequence, they excite magnons much more efficiently than phonons. At the frequency/field condition for mode X, the phonon mode is no longer pure phonon, but hybridized with magnons. However, as we discussed in the response to Reviewer #1, the longest propagating modes we observed might well be somewhat off the anti-crossing region. These phonons couple less strongly with magnons, and in this case, the “exciting magnons along the way” picture may be appropriate.

Note that if magnon relaxation frequency is still smaller than the resonance frequency shift of phonons due to the coupling, one may still talk about hybridization. If magnon relaxation is stronger, magnons excited by phonons are unable to come back to interact again with the phonons, preventing them from hybridization. To summarize, our answer to the question is that what we observe is a mixture of the hybrid modes and phonons leaking magnons away. We appreciate that it is an interesting conceptual question, but cannot offer quantitative analysis on it due to the limited experimental resolution and difficulty in systematic fitting of data by the theory.

Further, in the main text the authors write when discussing the hybridized state in line 177-178 “enhancement in the transmission spectra is observed” but in the supplementary material they also write in section VI. “[...] reflected upon the strong enhancement of the S_{22} signal which is not related to the magnon-phonon hybridization.” Could the authors please clarify?

Response:

We regret that this statement was confusing and blurring our main point. In the main text, we focus on the S_{21} (transmission) signal for BFO/LSMO samples, and we interpret the enhanced signal and decay length as arising from the magnon-phonon hybridization. In the supplementary, we also show the S_{22} (reflection) signal (now Fig. S13) in which we see a little similar signal enhancement near 45 degree. Since for reflection the propagation distance should not matter, the almost-flat dispersion at least partially explains the behaviours in Fig. S13b. We conclude that the magnon-phonon hybridization and the flat magnon dispersion both contribute to a strong mode X, particularly in the reflection spectra. We have now rephrased this discussion in the supplementary information as follows:

“This phenomenon may contribute to the strong enhancement of the S_{22} reflection signal (Fig. S13b, top row in Fig. S7) in addition to that of the magnon-phonon hybridization. However, the strong transmission S_{21} of mode X mainly results from the magnon-phonon hybridization since a flat magnon mode has zero group velocity. Instead, a magnon-phonon hybridized mode exhibits high group velocity and long life-time.”

It is known from cavity magnonics that the linewidths are changing towards the hybridization. Today, one distinguishes between coherent coupling with anticrossings as their hallmark and dissipative couplings with level crossings. It was shown that in the presence of dissipative coupling the linewidth of the hybrid state is decreased, and the amplitude increases. Furthermore, similar to some works in cavity magnonics (One tone approach from Harder et. al, PRL 2018 two tone approach, theory: Grigoryan et al. PRB 2018, experiment: Boventer et al., PRR 2020), the coupling strength becomes complex or purely imaginary as is the expression of the coupling strength from the authors in the supplementary material. I wonder if the authors have considered the possibility of such type of hybridization while seeing ‘only’ an enhancement in the spectra at the position of the conditions for the hybridization (Fig. S11) and if it could at least partially account for the observation of such ultra-low relaxation rate the authors reported. Could they please also comment on that?

Response:

This is a very insightful remark. Yes, we believe that the type of coupling at the crossing point may as well be level attraction. Nevertheless, from experimental results such as the newly added Fig. 3a, we cannot tell for sure whether it is anti-crossing or level attraction, simply because our measurement resolution is insufficient. We agree that the level attraction may result in line narrowing and enhancement of the signals as studied by previous works. Therefore, although we stay on the conservative side and put coupling type as anti-crossing in the simulation shown in the new Fig. 3b, we do not rule out the possibility of level-attraction type of coupling at the crossing point as shown in simulation results in the right figure below,

Note that the value of effective coupling $b_{\text{eff}} = 2 \times 10^7$ used here is way too large and would give an S_{21} simulation result inconsistent with the experimental data. Therefore, we have added a sentence in the later part of the main text to include this discussion as

“Although here we assume the coupling type to be *anti-crossing*, it may also be of *level-attraction* type [49] which may induce signal enhancement[50] and line narrowing[51] around the crossing point.”

[49] Harder, M. et al. Level attraction due to dissipative magnon-photon coupling. *Phys. Rev. Lett.* **121**, 137203 (2018).

[50] Grigoryan, V. L., Shen, K. & Xia, K. Synchronized spin-photon coupling in a microwave cavity. *Phys. Rev. B* **98**, 024406 (2018).

[51] Boventer, I. et al. Control of the coupling strength and linewidth of a cavity magnon-polariton. *Phys. Rev. Res.* **2**, 013154 (2020).

We also appreciate that the input-output theory perspective of Ref. [51] could offer a further insight into our experimental data. For instance, we could regard the CPW microwave mode as a dynamical entity, which couple with phonons via magnons. However, this analysis would require rewriting the theoretical part from scratch. Given that the present model already satisfactorily explains the data, we would like to leave it for a future study. Nonetheless, we revised Sec. VI in Supplementary to discuss anti-crossing and dissipative coupling.

What would be the lowest coupling strength possible to still see such long decay length?

Response:

If the dissipation is ignored, the anti-crossing theoretically occurs for any nonzero coupling. Thus, if the damping is zero and one has an infinite experimental resolution, there should always be a small frequency and field range in which the signal enhancement occurs. In reality, of course, the damping is nonzero, in which case the lowest value of coupling required is such that the resulting anti-crossing gap is greater than the linewidth. In fact, even if the coupling is not fully in a strong coupling regime, we still expect some sort of hybridization between magnons and phonons. Once the hybridization is formed, the phonon high velocity and long life-time will positively affect the magnon decay length. However, if the coupling is too weak, we doubt if we can still clearly observe the hybridized mode (mode X) in experiments. For example, when the magneto-elastic coupling is reduced to $b_{\text{eff}} = 2 \times 10^4$ in the simulation, as shown in the following figure, mode X vanishes with a propagation distance of 6 μm .

To be more precise, let us also point out that the simulation involves a fitting parameter c , which is taken $c = 6000$ for the main results and the figure here. This choice seems optimal for $b_{\text{eff}} = 2 \times 10^5$, but for other values of the coupling, one is free to adjust c to make the result look more like the

experiment. For weak couplings, one could indefinitely increase c to make the mode X visible. The honest situation is that we cannot really determine b_{eff} and c independently, but only their combination. In reality, c should be a material parameter, and a large c means a very sensitive piezoelectric effect. So there should be a minimal coupling strength required for observing the long decay length, but we are unable to pin it down quantitatively.

More discussion and relevant simulation results are shown in the revised Supplement Sec. VI.

In the following I only have minor comments.

a) What is the initial direction of the, for instance, out-of plane component, of the ferroelectric polarization state of the BFO layer? How does it affect the hybrid X mode if that state would be flipped, as it has been shown that the ferroelectric domain polarization state also can have an impact?

Response:

The initial direction of the out-of-plane component of the ferroelectric polarization is upward and we characterized with piezoelectric force microscopy (PFM), as shown in the following figure.

In our device with integrated CPWs or NSLs, the antennas are very fragile against the voltage applied by the PFM tips. Therefore, to study the spin-wave propagation while switching the electric polarization of BFO is not straight-forward in our experimental setup. However, we believe that further investigation of voltage-controlled spin-wave propagation in particular of the mode X would be very interesting, although it is beyond our current technique and out of the scope of the main focus of this work. Thanks to this helpful question from the reviewer, we became aware of a very recent work [Merbouche et al. *ACS Nano*, **15**, 9775-9781, (2021)] on voltage-controlled magnonic crystal demonstrated on relevant material system, and therefore we added one sentence in the outlook section to very briefly discuss about this point as

“In view of the multiferroic nature of BFO, future investigation may lead to voltage manipulation [52] of the magnon-phonon hybridized mode by switching the ferroelectric polarization.”

[52] Merbouche, H. et al. Voltage-controlled reconfigurable magnonic crystal at the sub-micrometer scale. *ACS Nano* **15**, 9775-9781 (2021).

b) The scale bars are only mentioned in the captions. I would suggest adding the scales rather to the image, helping to grasp the entire figure even faster.

Response:

We thank Reviewer #2 for this suggestion and have added the values of scale bars in all figures in the updated version of the main text as well as the supplementary information.

c) What is the orientation of the NGO substrate, 001 or 110, for instance?

Response:

The out-of-plane crystalline orientation of the NGO substrate used in this work is [001].

d) Line 258: The value of 29 GHz/T for the gyromagnetic ratio is a fit result and hence not 28 GHz/T?

Response:

Yes, the PPMS with phaseFMR module automatically yields a value of 28.5 GHz/T for the gyromagnetic ratio from fitting of the FMR measurement presented in Fig. S3. Therefore, we have used roughly 29 GHz/T in the simulation, as we can of course also use 28 GHz/T, which will not make much influence on the simulation results, we believe.

Recommendation:

In conclusion the observation of such a long decay length marks a significant advance for magnonics and beyond. The utilization of such hybridized mode opens new routes for the integration of magnonic devices to other architectures and opens new research directions based on such BFO/LSMO heterostructures as it can merge the functionalities of hybridized states and functional oxides together.

Thus, in principle, I can recommend this work for publication after minor revision in Nature Communications if the nature and its experimental demonstration of the hybridized state is further clarified.

Response:

We thank Reviewer #2 for his/her positive comment **“the observation of such a long decay length marks a significant advance for magnonics and beyond”** and the recommendation for publication.

Reviewer #3:

The paper presents experimental evidences of waves propagating in BFO/LSMO bilayer deposited on a NGO substrate. Two modes are evidenced. The slower one is the dipole exchange mode propagating perpendicularly to the magnetization direction in LSMO. The faster one propagates with a group velocity of 2.5 km/s. This latter one is interpreted as an acoustic wave propagating in BFO. The experimental results are nice but the interpretation is questionable (see the attached file). I suggest major revisions before publication.

Response:

We thank Reviewer #3 for having nicely summarized the manuscript.

The experimental results are nice but the interpretation is questionable because 2.5 km/s corresponds to shear wave velocity in LSMO [Phys. stat. sol. (a) 195, No. 2, 350–358 (2003)]. According to [Appl. Phys. Lett. 102, 182905 (2013)] the BFO shear elastic constant C_{66} is 89 GPa. Taking a BFO mass density of 8220 kg/m³, we obtain a shear wave velocity of 3.3 km/s. According to [Eur. Phys. J. B 94, 108 (2021)] the NGO shear modulus is 90 GPa. Taking a NGO mass density of 7.3 kg/m³, we obtain a shear wave velocity of 3.5 km/s. According to these estimations, the LSMO shear wave velocity is lower than those in BFO and NGO. This means that LSMO sandwiched between BFO and NGO is an acoustic wave guide. Consequently the observed wave propagating at 2.5 km/s is localized within LSMO. Coupling acoustic waves and dipole exchange waves within LSMO is more likely than coupling waves in different media BFO and LSMO.

Response:

We thank the reviewer for raising this question. First, if the coupling were between phonons and magnons within LSMO only, we should be able to observe long-distance propagation of mode X in pure LSMO sample. However, from Fig. S4b, we can see that mode X is almost invisible in the transmission spectra. This would suggest that the coupling with BFO is necessary to form the hybridized mode X. Second, we note that the pure phonons in BFO may have slightly higher group velocity than the observed mode in the experiments. As mentioned in the response to Reviewer #1, we measured the sound velocity for a pure BFO sample and obtained 3.2 km/s. This may result from its hybridization with magnons whose group velocity is much slower, below 1 km/s. This is in fact an indirect evidence that the observed long-distance mode has the nature of not pure phonons, but rather the magnon-phonon hybridized mode. Third, we also remark that hybridization between magnons and TA phonons should have a different dependence on the magnetic field direction than the hybridization with LA phonons.

We thank the reviewer for pointing out reference [Appl. Phys. Lett. 102, 182905 (2013)]. Based on their numerical model calculation, the phonon velocity in BFO can be estimated to be 3.3 km/s which is higher than the TA phonon velocity 2.2 km/s reported experimentally by neutron scattering in Ref. [33]. Furthermore, 550 m/s was obtained for surface mode by a transmission experiment in Ref. [25]. If the surface mode is a Rayleigh wave, which is quite probable here, its velocity is only up to around 10% smaller than the TA velocity, which suggests a value significantly lower than 3 km/s. Generally, therefore, the numerical results of [Appl. Phys. Lett. 102, 182905 (2013)] seem to overestimate the elastic constants in comparison with the experimental data of [25,33]. Still, all these results point to LA sound velocity of order 3 km/s for BFO, which seems at least consistent with our interpretation considering the uncertainties in all the experiments as well as in the numerics. We have now included these discussions in the main text as

“The magnon-phonon coupling within LSMO cannot account for the hybridized mode X in view of the fact that mode X vanishes in the transmission spectra on a bare LSMO film without BFO (see Supplementary Fig. 4). A neutron scattering experiment [33] reported the velocity of longitudinal phonon mode to be 2.6 km s^{-1} , which is fairly close to the value (3.2 km s^{-1}) estimated from our experiments on a pure BFO sample (see Supplementary Fig. 14). Therefore, we assign mode X to the BFO phonon mode hybridized with LSMO magnon mode. The velocity of mode X (2.5 km s^{-1}) is slightly lower than that of phonons in pure BFO [33,46] due to the hybridization with the slow magnon mode.”

[46] Dong, H., Chen, C., Wang, S., Duan, W. & Li, J. Elastic properties of tetragonal BiFeO₃ from first-principles calculations. *Appl. Phys. Lett.* **102**, 182905 (2013).

Concerning the model, could the authors explain why the exchange bias field has the same in-plane and out-of-plane contributions and why the in-plane contribution depends not on magnetization direction?

Response:

We regret that this caused some confusions. The constant B was introduced as a convenient fitting parameter since the precise modelling of all the magnetic free energy contributions in such a complicated heterostructure is impossible. We are not sure about the origin of B, but one possibility could be that the antiferromagnetic Néel vector rotates as the in-plane magnetic field is rotated (possibly by minimising the magneto-static energy arising from the weak ferromagnetism of BFO), in which case the exchange bias should appear isotropic for the LSMO magnetization. However, this is just one possibility and we do not have any supporting evidence, so that we have removed this remark and simply introduced B as a fitting parameter in the revised manuscript.

Response to the second referee reports

We appreciate the additional comments and suggestions from Reviewers #1 and #2. We also thank Reviewer#3 for his/her full support on publishing our manuscript in *Nature Communications*. Following their useful suggestions, we have made more theoretical analysis and conducted additional BLS measurements. Below we provide the detailed response to the referee comments in a point-by-point manner.

Reviewer #1 (Remarks to the Author):

In the revised manuscript, the authors properly answered most of my questions/comments, some of them are still not very clear yet. I can recommend the manuscript for publication in *Nature Communications* after clearly addressing the rest problems.

(1) The authors experimentally demonstrate a long propagation length of the acoustic wave in BFO using IDT (Fig. S14). However, the Q-factor of the mode at $f \approx 1.8$ and 5 GHz is low (<50). Since the frequency and linewidth of an acoustic wave is related to the IDT geometry, can the authors provide more information on the IDT such as electrode width, period, and number of electrodes? Does the mode of $f \approx 5$ GHz correspond to a harmonic mode? Can the authors also comment on the low-Q?

Response:

We thank Reviewer#1 for his/her insightful question. The number of IDT electrodes is 20, the electrode width of is 600 nm, and the period is 4 μm . Although one may roughly read them out from Fig. S14a, we have now provided these values explicitly in the supplementary information to facilitate the readers. Based on these parameters, we have performed the Fourier transformation to calculate the wavevector excitation of the IDT, which has been added as new Fig. S14b. In the wavevector distribution, there are three peaks corresponding to three modes in experiments. The mode around 5 GHz is attributed to the second harmonic mode.

As for the Q factor, if one directly takes the linewidth of each peak to estimate the Q factor, it would be very low. Nevertheless, the main contribution of the linewidth is given rise by the broad wavevector excitation of the IDTs as shown in Fig. S14b. If we may exclude the contribution from the wavevector broadening, the Q factor originating from the damping of the phonons should be very high.

(2) In BFO/LSMO, long decay length of the mode X is attributed to the coupling of the magnon to the phonon, i.e., the low-loss acoustic wave carrying the magnon propagates over 1 mm. Going back to my question about the coupling for $f > 6$ GHz (Question 4), the authors argued the coupling of magnons at high frequency to the high-order phonon modes e.g., the k4 mode, however, the signal of the mode X seems to be stronger with the magnetic field increasing (see Fig. S18b and d). As the k4 phonon mode is weakly excited by the CPW, one expects to observe the weaker mode X at the higher frequency. By comparing the data in Fig. S13d and Fig. S18, the phonon intensity measured by the CPW seems to be different, i.e., the phonon modes are not observed in Fig. S18. Can the authors comment on those? In addition, can the authors evaluate theoretically the coupling strength extracted from the anti-crossing gap for different phonon modes, e.g., k1 - k6 (see main Fig. 3b). In the experiment, thermal phonons of high frequency may play a role in the coupling. This may explain why the coupling occurs in a wide frequency range.

Response:

We thank Reviewer#1 for his/her insightful observation and great suggestions. Indeed, the higher order k4 mode should be even weaker than k1, k2 and k3 modes. However, the N sample series have in general much broader excitation spectrum than the Z series, as can be seen from the lack of k1, k2, k3 peaks in Fig. 18b. Therefore, we do not expect a clear peak structure suggested by the Fourier transform. Instead, the intensity should be more continuously distributed for the N series. The difference between Z and N series could be due to bad contact between the CPW and substrate, although we cannot be sure.

The coupling strength in the theoretical model is constant b_{eff} independent of k. If we take the thin film limit $kt_{\text{FM}} = 0$ for simplicity, the coupling strength is related to the anti-crossing gap $\Delta\omega$ by

$$\frac{\Delta\omega}{c_s k} = 4 \sqrt{\frac{b_{\text{eff}}^2 \sin^2 2(\beta - \theta_M)}{\rho c_s^2 \times \mu_0 H M_s}}.$$

This formula has a simple meaning: Note ρc_s^2 and $\mu_0 H M_s$ are the characteristic energy densities of phonons and magnons respectively and b_{eff} is the coupling strength in the unit of energy density. Therefore, the ratio between the gap and the original resonance frequency $c_s k$ is equal to the ratio between the coupling energy and the geometric mean of the energy of phonons and magnons. Estimating the gap for $b_{\text{eff}} = 2 \times 10^5$ yields $\Delta\omega \approx 10\text{MHz}$, again roughly independent of k, in agreement with Fig. S8.

We agree with the reviewer that the observed modes at higher frequency with a wide frequency range may be due to the effect from thermal phonons and have now added one sentence around the Fig. S18 to further discuss this issue as

“It is worth noting that the thermal phonons excited in the experiments might play a role in the formation of the observed magnon-phonon hybridization mode with wide frequency range and relative higher frequency.”

(3) In BLS measurements, the authors mention that the decay of the mode X in the 2D spatial map, particularly in the first few micrometers, is primarily determined by the magnon decay length. Can the authors present a 2D image with a large y range where the hybridized mode X can be visualized?

Response:

Following the reviewer’s comment, we measured a 2D map of mode X at a larger y range. The piezoscanner allowed us to cover up to 60 micrometers in y-direction with high spatial resolution. Thereby we have extended the range of y-coordinates by a factor of 20 compared to the previous data. The stripline antenna is located at $y = 0$. The BLS intensity map from $35 \mu\text{m}$ to $58 \mu\text{m}$ in y-direction is presented in the newly added Fig. S20. As mentioned by the reviewer, BLS signals at the first few micrometers is predominated by the pure magnon mode with a short decay length of $\lambda_m = 5 \mu\text{m}$ as extracted from Fig. 5 of the main text. At $y = 35 \mu\text{m}$, the pure magnon mode with $\lambda_m = 5 \mu\text{m}$ should already vanish. Thus, the remaining signals consist mainly the magnon-phonon hybridized mode which decays much slower amid its weak signal strength. Due to the limited range of the piezoscanner we can map the mode X by BLS measurements up to $58 \mu\text{m}$ only. The inhomogeneity of the signal is attributed to the roughness of the antenna.

(4) The title of the manuscript is “long magnon decay length in BiFeO₃/La_{0.67}Sr_{0.33}MnO₃ heterostructures”. Since long decay length doesn’t apply for pure magnons in BFO/LSMO, to not be confused, the author may think about a different title like “Long decay length of magnon-polarons in BiFeO₃/La_{0.67}Sr_{0.33}MnO₃ heterostructures”.

Response:

We find this suggestion quite reasonable, and have accordingly modified the title as “*Long decay length of magnon-polarons in BiFeO₃/La_{0.67}Sr_{0.33}MnO₃ heterostructures*”

Reviewer #2 (Remarks to the Author):

I would like to thank the authors for their detailed response on the other reviewers and my own remarks to their manuscript on the long magnon decay length in BFO/LSMO heterostructures.

The authors have mostly answered my concerns and together with the other reviewers comments I also believe that the manuscript has much improved. However, there is one main point remaining which still needs to be clarified in my view (Please also have a look on my comments in the attached file).

The experimental data is well presented and definitely interesting. The authors have greatly improved the presentation and discussion of these in the main part of the manuscript.

Response:

We thank Reviewer#2 for supportive comment on our work and also our previous efforts on improving the manuscript. Below we response to his/her questions in a point-to-point manner.

However, to me the possible underlying physical mechanisms for that observation of such a long decay length- are listed but its not clear what is the dominating one governing the propagation.

Correspondingly (also see my comments) I don’t consider the answer to my “excitation of magnons in the fly” sufficient to explain the main source of the long decay length (also in combination with the 2D map provided now by the authors)/ and the one to anticrossings/level attraction. Could the authors please further comment on that?

Response:

We thank Reviewer#2 for the important question on the mechanism of the observed long-distance mode. In our work, the observed mode X with long decay length is considered to originate from the “off-crossing-point” coupling between magnon and phonon, i.e. the phonon mode slightly hybridized with magnons as indicated by the green circle in Fig. 3b. In this case, such hybridized mode can retain the high velocity and long lifetime of phonons, while carries the magnon information.

Reviewer#2's comments and suggestions in the attached pdf file:

Can the magnon/phonon for the data shown in the work content be estimated?

Response:

We attribute the horizontal mode in Fig. 3a to the k_3 phonon mode excited by the antenna. From the k value of the k_3 mode is found to be slightly lower than the value at the crossing point. The frequency observed for mode X (Fig. 3a) is around 7 GHz, which is also slightly lower than the crossing point in Fig. 3b.

I don't fully get the answer to Reviewers#1 question regarding the enhancement of the decay length. This looks to me like a more detailed explanation of the mechanism especially regarding the coupling.

Response:

In simple terms, the magnons inherit the long decay length of the phonons via magneto-elastic coupling. If the magnon-phonon hybridization is 50-50, the enhancement of decay length would be quite limited. But if the hybrid mode is dominated by phonons, say 10-90, it travels essentially like phonons and still carries magnetic signal.

In a usual magnon transmission experiment, it could be difficult to observe 10-90 modes because both excitation and detection are magnetic so that the signal would be reduced by 99%. In our case, however, the multiferroic nature of the system helps and the detection side seems dominated by the piezoelectricity, which is a very sensitive effect.

Panel (a) is simulation or why is the SNR so different to the one shown in panel (d)

Response:

Panel (a) is the transmission spectra at the propagation distance of 12 μm , while for panel (d) the propagation distance is 940 μm , and thus the signal is much weaker and the SNR is worse.

On what basis is this number estimated? Available material parameters or some other parameters?

Response:

As we demonstrated in Fig. S9, this number is optimal for reproducing the observed spectrum by the numerical calculation. The material value of the magneto-elastic coupling b does not seem to be available in literature and in general difficult to determine experimentally. In addition, our experiment is sensitive to only b_{eff} , which involves b and another unknown model parameter $t_{\text{interface}}$, as explained in the previous reply. Thus knowing the material value from our measurement is impossible even as a matter of principle.

maybe add a sentence to the discussion/outlook of the manuscript?

Response:

We thank the reviewer for this helpful suggestion and have therefore added a sentence in the discussion part as:

“The conventional DE spin-wave mode (mode Y) shows clear nonreciprocity, while the magnon-polaron mode (mode X) is rather reciprocal. The different behaviour in spin-wave nonreciprocity may be potentially useful for spin-wave computing and signal processing [54]. The reciprocal property of mode X may be further studied in future works.”

[54] G. Csaba, A. Papp & W. Porod, Perspectives of using spin waves for computing and signal processing. *Phys. Lett. A* **381**, 1471-1476 (2017).

If there is a reference to the strong coupling regime a reference to the values of the phonon, magnon linewidth and the coupling strength would be most helpful as it allows to quickly grasp the condition $g \gg \kappa_{\text{phonon}}$, κ_{magnon} for entering the strong coupling regime.

Response:

We have added a reference of (Godejohann, F. et al. Magnon polaron formed by selectively coupled coherent magnon and phonon modes of a surface patterned ferromagnet. *Phys. Rev. B* **102**, 144438 (2020).) to show the condition $g \gg \kappa_{\text{phonon}}$, κ_{magnon} for entering the strong coupling regime. In this reference, the magnon-phonon coupling strength g is estimated to be 0.2 GHz and dissipation rates for magnon and phonon are $\kappa_{\text{magnon}} = 0.17$ GHz and $\kappa_{\text{phonon}} = 0.03$ GHz, reaching the strong coupling regime. In our work, we do not claim to enter the strong coupling regime.

Referring to my comment in my first review this looks like "excitation on the way". The mode X decays but revives then at 10 μm and then goes down again at 18 μm ? I know these measurements take long but did the authors take a similar image far away from the antenna (close to 1mm) and check if the spatial of the X mode looked similar?

Response:

We are really sorry for the misleading 2D spatial map in Fig. S19, where the propagation direction of the magnon-phonon hybridized mode is along y direction rather than the x direction. The gold strip in the middle of the map is the antenna used for magnon excitation. We have revised the corresponding denotation in the supplementary material for a clearer explanation. From the 2D map we displayed in Fig. S19, the propagation range is within 3 μm , and from Fig. 5 we know at such a short distance the measured signal intensity is dominated by the pure magnon mode.

Along y axis, the observed non-uniform excitation of mode X in the map is due to the

roughness of the antenna, the width of the antenna is not perfectly uniform along the antenna, thus the intensity of the excited magnon along the antenna also displays a non-uniform distribution.

We have now added another 2D map of mode X measured by BLS to the supplementary material as the new Fig. S20. As mentioned above in the response to reviewer #1, we are limited to $\sim 60 \mu\text{m}$ from the antenna due to the piezoscanner in our BLS setup. We could not provide a 2D map as far as 1 mm from the antenna as the manual displacement and positioning of the BLS sample holder would introduce uncertainties, because the antenna is no longer in the field of view and the precise alignment of magnetic field direction with respect to the antenna as well as distance from the antenna are uncertain. Nevertheless, as stated previously, the BLS data measured in the region from $y = 35 \mu\text{m}$ to $58 \mu\text{m}$ already show a very different decay behavior compared to the first few micrometers.

Could you please provide some numbers?

As shown in Fig. S10 and in the reply to Reviewer #1 above, the anti-crossing gap for $b_{\text{eff}} = 2 \times 10^5$ is of order 10 MHz. Our Gilbert damping is 1.2×10^{-3} , and for $\omega = 6\text{GHz}$, it corresponds to the relaxation rate of around 7 MHz. Thus the coupling seems marginally better than the intrinsic damping of magnons. We have added this estimate in the main text (in the paragraph before Fig. 4). On the other hand, the extrinsic damping (linewidth) cannot be quantified systematically. If we estimate it from the width of phonon peaks in Fig. S13d, it would be around 1GHz. But the magnon-phonon signal (the bright line roughly linear in H in Fig. S13d) is clearly much narrower. In any case, the extrinsic width appears larger than 10 MHz.

In addition, I would like to thank the authors for adding this figure to the manuscript, but I still have some question:

The data looks like a "half cut" anticrossing comparing the left (<resonance field) and right side. So I wonder if there was some kind of background correction which may have added some unwanted artefacts? Additionally what parameters is the colorbar? [S11], [S21]? Its only in the caption in the revised manuscript

Response:

We thank Reviewer#2 for this insightful question. Indeed, there is a background correction in the spectra of Fig. 3a in order to minimize the large background noise, where the spectra is subtracted by reference spectrum at 0 mT magnetic field. This is why the spectra look like "half-cut" on the left side. If the spectrum at 250 mT as reference is taken as reference, we attain the spectra below showing "half-cut" on the right side,

The data presented in Fig. 3a is the imaginary part of the transmission spectra S21 after subtraction by the spectrum at 0 mT. The color bar denotes the signal strength of the imaginary S21 in arbitrary units. We have added this information in the caption of Fig. 3a following the reviewer's suggestion.

But in the 2D map of BLS data it could be seen that the X mode first experiences a fast decay due to the magnons (in line with the predominant excitation of magnons at the antennae), but what about propagation distances beyond that?

Response:

The antenna is more efficient on the magnon excitation at the excitation side, that's why we could still observe a fast decay due to the magnons. From Fig. 5 we could see that beyond the propagation distance of 5 μm , the decay rate of the hybridized mode turns to be smaller than the pure magnon mode but still experience an exponential decay.

As mentioned above, measuring propagation distances beyond $\sim 60 \mu\text{m}$ from the antenna was not performed due to limitations of the piezoscanner and concomitant uncertainties. While from the 2D map of BLS in a longer distance from $y=35 \mu\text{m}$ to $58 \mu\text{m}$, we see that the magnon-phonon hybridized mode decays with a much larger decay length than λ_m , where pure magnon modes should already vanish, and the remaining BLS signal arises from the magnon-phonon hybridized mode.

Is it the linewidth which is meant by relaxation frequency?

Response:

Yes, it is the frequency linewidth which is meant by relaxation frequency.

I partly agree on the author's statements and would like to thank them for further pointing it out. However, unfortunately, I am not convinced. The paper reports on the long magnon decay length, although the experimental data is nice I still think the physics/interpretation for the reason of such observation is not fully clear to me what is the main drive behind the

observation of such long decay length.

Response:

We thank the reviewer for his/her strong interest in the fundamental mechanism behind our observation. Essentially the same question was raised above, and responding to the strong interest from the reviewer, please let us reiterate our consistent answer to it.

In BFO, there exist phonon modes that can travel more than 1mm, as confirmed in Fig. S14, from which the phonon decay length can be estimated to be about 1.7 mm. We attribute the longest travelling magnons in our experiment to those that are weakly hybridized with the phonons in BFO. The magnon-polaron dominated by the phonon contribution can have the group velocity and damping of the same order as those of pure phonons so that they can travel as much distance as phonons, which is much longer than pure magnons. These “off-crossing” modes still carry distinctively magnetic signal, i.e. a characteristic dependence on the magnetic field direction, which we demonstrated by our theoretical calculations.

In this process, the multiferroic heterostructure BFO/LSMO plays an important role in making this “off-crossing” mode detectable. Without the piezoelectric signal arising from the ferroelectric BFO, it would be difficult to see the phonon-dominated magnon-polaron as the magnetic signal is diminished accordingly. Here we are seeing the electric signal from the phonon part of the same magnon-polaron via piezoelectricity.

And if the same value of the main part of the paper is used there is nothing to be seen in the spectrum for level attraction? Usually, in experiments, the level attraction spectrum is much less separated.

Response:

We have added the calculated spectra for level attraction by using the magneto-elastic coupling strength of $b_{\text{eff}} = 2 \times 10^5$ in Fig. S10. It seems that by changing the coupling type from anti-crossing to level attraction, the transmission spectra of mode X is still similar, because when the coupling is weak, the coupling type does not make much difference in the spectra as in Fig. S10. Therefore, based on the experimental results shown in Fig. 3a, it is very hard to distinguish anticrossing from level attraction because in our case the coupling between magnon and phonon may not be in the strong coupling regime.

Whether the coupling is coherent (repulsion) or dissipative (attraction) has a great impact on linewidths (see eg recent reviews on cavity magnonics/cavity magnon polaritons).

Consequently, if the linewidth is smaller the magnon-phonon state will also propagate longer.

On what basis is the assumption to "stay on the conservative side and use anti-crossings"?

Maybe the one spectrum shown in Figure 3a of the main?

Response:

We very much appreciate that the reviewer brought it into our attention, and totally agree that the coupling type has a great impact on linewidths. Dissipative coupling can reduce the linewidth significantly under some circumstances. We simply do not have enough supporting evidence for that happening in our samples, however.

As we answered in the previous question, the coupling strength that makes simulation results consistent with the observed spectrum is too small to allow us to distinguish between ordinary and dissipative couplings. Please note that $b_{\text{eff}} = 2 \times 10^5$ corresponds to 10 MHz in frequency, and the dissipative coupling of this strength can reduce the linewidth in the frequency region of 10MHz around the crossing point, but not beyond. We clearly see the long propagating signals in the broad frequency range of order 100 MHz (Fig. 2, S17). In this region, the magnon-phonon hybridization is weak, and the coupling type does not make a big difference.

Dissipative coupling remains a possible explanation for the long decay length in our experiment. However, please let us stress that there is no known physical mechanism that gives rise to a dissipative magnon-phonon coupling. The situation is very different from the FMR mode, for which a dissipative coupling is known to arise from Lenz effect with a cavity photon, or from Slonczewski torque with another FMR mode. Since we do not have any direct evidence for a dissipative magnon-phonon coupling, we are unable to claim that it exists in our system just because of the long magnon decay length.

In contrast, the ordinary magnon-phonon coupling definitely exists. Our scenario attributes the long magnon decay length to the combination of a long phonon decay length with a weak magnon-phonon hybridization. We have collected several pieces of evidence that support our claim, as detailed above and in the manuscript, and believe that they are sufficiently strong for making our case. Of course, further studies will be needed to consolidate our interpretation.

To acknowledge that we are unable to exclude dissipative coupling and it could contribute to the long decay length, we have added the following sentence in the main text.

“Although here we assume the coupling type to be anti-crossing, we admit that it is hard to

distinguish anti-crossing from level-attraction type [50,51,52] which may induce signal enhancement [53] and line narrowing [54] around the crossing point, especially with intermediate coupling strength (Supplementary Fig. 10).”

[50] Harder, M. et al. Level attraction due to dissipative magnon-photon coupling. *Phys. Rev. Lett.* **121**, 137203 (2018).

[51] Rameshti, B. Z. et al. Cavity magnonics. Preprint at <https://arxiv.org/abs/2106.09312> (2021).

[52] Harder, M., Yao, B. M., Gui, Y. S. & Hu, C.-M. Coherent and dissipative cavity magnonics. *J. Appl. Phys.* **129**, 201101 (2021).

[53] Grigoryan, V. L., Shen, K. & Xia, K. Synchronized spin-photon coupling in a microwave cavity. *Phys. Rev. B* **98**, 024406 (2018).

[54] Boventer, I. et al. Control of the coupling strength and linewidth of a cavity magnon-polariton. *Phys. Rev. Res.* **2**, 013154 (2020).

Based on this, I still could recommend the manuscript for publication in *Nature Communications*, if this answer is answered more clearly as it is indeed an important question in (hybrid) magnonics.

Yours sincerely,
Reviewer

Response:

We thank the Reviewer#2 for his/her positive support on publishing our work in *Nature Communications*, and more importantly for providing us detailed and insightful comments and suggestions, which we believe have helped us a lot in improving the manuscript.

Reviewer #3 (Remarks to the Author):

the novelty of the experimental data and the effort carried out for addressing the issues raised in the original version make the present version suitable for publication in *Nature Communication*

Response:

We greatly appreciate Reviewer#3's positive comments and recommendation for publication.

REVIEWERS' COMMENTS

Reviewer #1 (Remarks to the Author):

The authors answered my questions properly and included my suggestions into the revised manuscript. Therefore, I recommend the current version for publication in Nature Communications.

Reviewer #2 (Remarks to the Author):

To whom it may concern,

I would like to thank the authors again for their efforts to answer all questions from myself and the other reviewers.

I think my previous doubts have been sufficiently answered- although I still believe that further investigations are necessary to further explore this interesting phenomena in such heterostructures. However, these are outside of the scope of this work and it has been pointed out now sufficiently in the manuscript.

Thus, I can now recommend this work as it is for publication in Nature Communications.

Yours sincerely,

Reviewer

Response to referee reports

Reviewer #1 (Remarks to the Author):

The authors answered my questions properly and included my suggestions into the revised manuscript. Therefore, I recommend the current version for publication in Nature Communications.

Response:

We thank Reviewer #1 for his/her full support and recommendation for publication in Nature Communications.

Reviewer #2 (Remarks to the Author):

To whom it may concern,

I would like to thank the authors again for their efforts to answer all questions from myself and the other reviewers.

I think my previous doubts have been sufficiently answered- although I still believe that further investigations are necessary to further explore this interesting phenomena in such heterostructures. However, these are outside of the scope of this work and it has been pointed out now sufficiently in the manuscript.

Thus, I can now recommend this work as it is for publication in Nature Communications.

Response:

We appreciate the detailed suggestions from Reviewer #2 throughout the review process, which have considerably improved our manuscript. Meanwhile, we completely agree with him/her that further investigation on relevant phenomena in these multiferroic heterostructures would be interesting and may reveal more fascinating physics. Indeed, we plan to work further towards this direction thanks to the inspiration from the reviewer.